

# Planktonic foraminiferal assemblages as tracers of paleoceanographic changes within the Northern Benguela current system since the Early Pleistocene

Arianna V. Del Gaudio[1], Aaron Avery[2], Gerald Auer[1], Werner E. Piller[1], Walter Kurz[1]

[1]Department of Earth Sciences (Geology and Paleontology), NAWI Graz Geocenter, University of Graz, Graz, 8010, Austria
[2]School of Geosciences, University of South Florida, Tampa, FL, 33620, USA

*Correspondence to*: Arianna V. Del Gaudio (arianna.del-gaudio@uni-graz.at)

**Abstract.** The Benguela Upwelling System (BUS), located in the southeastern Atlantic Ocean, represents one of the world's most productive regions. This system is delimited to the south by the Agulhas retroflection region. The northern boundary of

the BUS is, instead, represented by the Angola Benguela Front (ABF), which is a thermal feature separating warm waters of the Angola Basin (including the South Atlantic Central Waters; SACW) from the cooler Benguela Oceanic Current (BOC). We performed statistical analyses on planktonic foraminiferal assemblages in 94 samples from Holes U1575A and U1576A, cored during International Ocean Discovery Program (IODP) Expedition 391. Drilled sites are located along the Tristan-Gough-Walvis Ridge (TGW) seamount track in the northern sector of the BUS (offshore the Namibian continental margin).

The analyzed stratigraphic intervals span the Early-Late Pleistocene, marked by the Early-Middle Pleistocene Transition (EMPT; 1.40-0.40 Myr), during which important glacial-interglacial sea surface temperature (SST) variabilities occurred. This work provides novel insights on the local paleoceanographic evolution of the northern BUS and associated thermocline variability based on the ecological significance of the foraminiferal assemblages. Specifically, variations in the assemblage content allowed to characterize the different water masses (BOC, SACW, Agulhas waters) and reconstruct their interactions

during the Quaternary. The interplay of the previously mentioned water masses induced perturbations in the BUS (ABF latitudinal shifts and input of tropical waters from the Agulhas retroflection region). Furthermore, we investigated the possible link between changes in the paleoceanographic conditions and climatic events (e.g., Benguela Niño/Niña-like phases and deglaciation stages) recorded since the EMPT.

## 1 Introduction

The Benguela Upwelling System (BUS), in the southeastern Atlantic Ocean, is known as one of the key temperate productive regions on Earth (Giraudeau, 1992; Little et al., 1997; Petrick et al., 2018) since the Middle Miocene (Diester-Haass, 1988). This area is, in fact, subjected to strong upwelling episodes, during which cold and nutrient-enriched subsurface waters rise to the surface along the southeastern coast of the African continent (Giraudeau, 1992; Little et al., 1997; Ufkes and Kroon, 2009; Rouault and Tomety, 2022). Interestingly, the BUS is also influenced by the ingressions of warm Indian Ocean waters (so-





called Agulhas leakage) via Agulhas eddies (Bé and Duplessy, 1976; Fine et al., 1988; Caley et al., 2012), as well as by the subsurface South Atlantic Central Water (SACW) from the Angola Basin (Mohrholz et al., 2008; Ufkes and Kroon, 2012). The meridional thermal front, which forms in the convergence zone between the warm subsurface waters and the cold Benguela waters, is named Angola-Benguela Front (ABF; Mohrholz et al., 2008; Kopte et al., 2017).

Modern reconstructions (e.g., Gammelsrød et al., 1998; Rouault et al., 2007) of sea surface temperatures (SSTs) within the

BUS, also reveal the existence of a strong interannual SST variability which induce severe warm (cold) events along the Angola-Namibia cost known as Benguela Niños (Benguela Niñas) events (Shannon et al., 1986; Imbol Koungue et al., 2019). These climatic phenomena severely affect the paleoceanographic conditions in the BUS (e.g., the position of the ABF; Walter, 1937; Boyd et al., 1987; Shannon and Nelson, 1996). Several studies suggested the occurrence of Benguela Niño/Niña events during the Pliocene-Pleistocene time intervals (e.g., Ufkes and Kroon, 2012; Rosell-Melé et al., 2014). Particularly, since the

onset of the Early-Middle Pleistocene Transition (EMPT), a strong glacial-interglacial sea surface temperature (SST) variability occurred, possibly promoting the development of Benguela Niño/Niña conditions (Herbert, 2023).

In this study, we analyzed planktonic foraminiferal assemblages during and after the EMPT to investigate the paleoceanographic history of the most distal sector of the BUS offshore the South African continental margin. Previous studies (e.g., Little et al., 1997; Giraudeau, 1993; West et al., 2004) mainly analyzed planktonic foraminifera assemblages in younger

sediment cores close to the continental margin, thus principally focusing on the coastal upwelling system. Moreover, Ufkes and Kroon (2012) investigated the palaeoecological conditions of the BUS in the last 1.1 Ma, not covering the whole EMPT interval. IODP Expedition 391 Sites U1575 and U1576 were drilled along the Tristan-Gough-Walvis Ridge (TGW) seamount track in the northern Benguela region, under the influence of the Benguela Offshore Current (BOC) and close to the ABF (Fig. 1 and 2). The location of the sites is ideal to detect regional changes in the paleoceanographic conditions within the distal part

of the BUS and to explore the interaction of the BOC with the Angola Basin and Indian Ocean water masses. Furthermore, the sites were cored in different sectors of the South Atlantic Gyre System (SAGS), with Hole U1575A located near the continental margin and Hole U1576A situated in a more southern position and closer to the center of the gyre (Fig. 2). This allowed to infer variations of the paleoceanographic conditions in different parts of the SAGS.

The major objectives of our study were to (1) characterize the BOC, the SACW and the Agulhas leakage in terms of planktonic

foraminiferal assemblage compositions; (2) examine the interaction of the BOC with the warm waters from the Angola Basin (which affects the ABF position) and the Indian Ocean waters (through Agulhas leakage) during the Quaternary; (3) detect the response of the regional thermocline to the variations of the paleoecological conditions in the area; (4) investigate the link between changes in the ABF position as well as in the influx of the Agulhas waters to climatic conditions (e.g. Benguela Niño/Niña-like events and deglaciation phases), which were previously documented to occur since the onset of the EMPT.






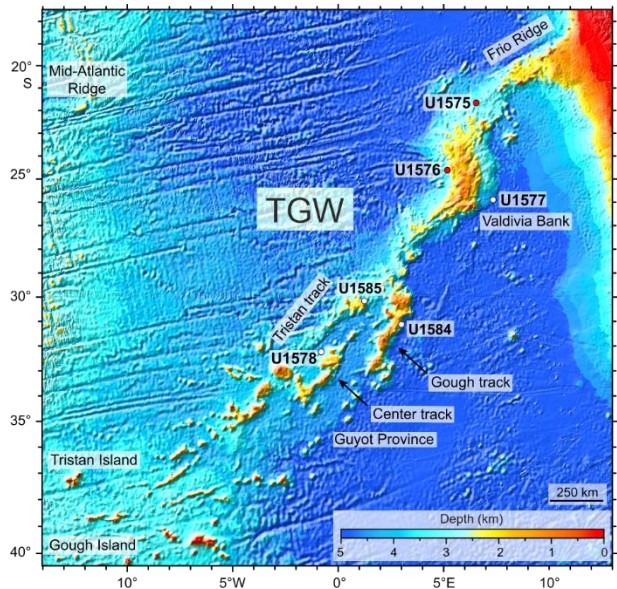

**Fig. 1:** Bathymetric map of the Tristan-Gough-Walvis Ridge (TGW) volcanic chain and its main seamount tracks (modified from Sager et al., 2023). Red dots=IODP sites used for this study.

## 1.1 Geological setting


The Tristan-Gough-Walvis Ridge (TGW) track is a volcanic chain in the eastern South Atlantic Ocean between 18°S and 32°S (Humphris and Thompson, 1982; Sager et al., 2022; Fig. 1). It extends for 3100 kilometers (km) in length from Tristan da Cunha and the Gough islands to the Namibia coast (Cabo Frio) in southwest Africa (Connary, 1972; Sager et al., 2020, 2022). Additionally, the ridge complex separates the Cape Basin in the south from the Angola Basin in the north (Shaffer, 1984). The

TGW shows a complex morphology (Connary, 1972; Humphris and Thompson, 1982; Thoram et al., 2023), as it is formed by three seamounts chains (Tristan Track, Central Track, and Gough Track) exhibiting different Pb isotopic compositions (Hoernle et al., 2000; Werner et al., 2003; Hoernle et al., 2015; Homrighausen et al., 2019; Sager et al., 2020). The TGW also comprises a continuous ridge (Frio Ridge), an oceanic plateau (Valdivia Bank), guyots, and scattered seamounts (Sager et al., 2020). The formation of TGW started in the Early Cretaceous (~132 Ma ago), and it is related to the initial rifting of the South

Atlantic Ocean. Some researchers (Wilson, 1965; Morgan, 1971; Detrick and Watts, 1979) suggested that the TGW was formed due to the lithospheric plate movement above a fixed hotspot. However, more recent studies (Fairhead and Wilson, 2005; Foulger, 2007) proposed that its origination could be linked to non-hotspot-related volcanism, along shear zones (Sager et al., 2022).

Sites drilled along the TGW during IODP Expedition 391 (U1575, U1576, U1577, U1578) recovered Upper Cretaceous

(Campanian-Maastrichtian) to Upper Pleistocene sedimentary sequences (Sager et al., 2022). Main lithologies include white



to pale brown calcareous nannofossil-planktonic foraminifera oozes, as well as brown-pink and light green to gray nannofossil-foraminifera chalks, seldomly interbedded with light to dark gray tephra layers.

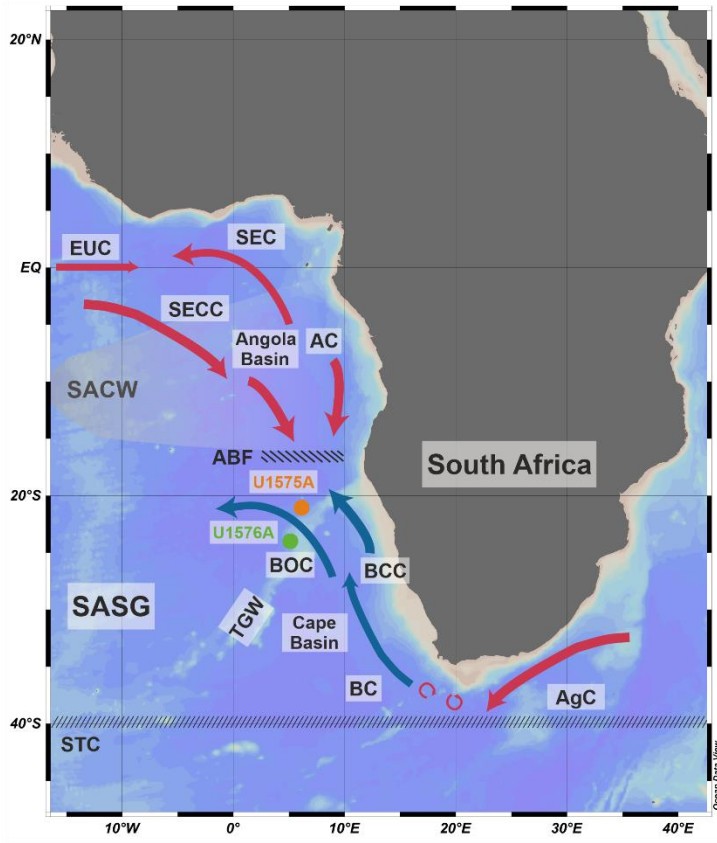

**Fig. 2:** Oceanographic setting of the studied area (adapted from Ufkes and Kroon 2012). The map was created using Ocean Data View
(Schlitzer, 2021). Orange and green dots=location of Holes U1575A and U1576A. Red arrows=warm water currents; blue arrows=cold
water currents; circular arrows=Agulhas eddies. Oceanic features and the names of the water currents are abbreviated as follows:
ABF=Angola-Benguela Front; AC=Angola Current; AgC=Agulhas Current; BC=Benguela Current; BCC=Benguela Coastal Current;
BOC=Benguela Oceanic Current; EUC=Equatorial Under Current; SACW=South Atlantic Central Water; SASG=South Atlantic
Subtropical Gyre; SEC= South Equatorial Current; SECC=South Equatorial Counter Current; STC=Subtropical convergence;
TGW=Tristan-Gough-Walvis Ridge.

## 1.2 Oceanographic setting

Holes U1575A and U1576A are situated in the eastern sector of the South Atlantic Subtropical Gyre (SASG; Fig. 2). The
SASG is a wind-driven, counterclockwise flow, which is responsible for the redistribution of energy between low and high
latitudes, thus controlling the global climate system (Gordon, 1973; Talley, 2003; Drouin et al., 2020; Pinho et al., 2021). It
roughly extends between 45°S–15°S and 55°W–10°E (Drouin et al., 2020). Four major water masses define the SASG, with





the Benguela Current (BC) representing the eastern sector of the gyre (Stramma and Peterson, 1990; Garzoli and Gordon, 1996; Stramma and England, 1999; Fig. 2).

The Benguela Upwelling System (BUS), located along the southern Africa continental margin, is known as one of the most productive regions in the oceans (Rouault and Tomety, 2022) in which the southerly trade winds induce upwelling of cold, nutrient-rich subsurface waters along the western side of South Africa and Namibia (Giraudeau, 1992; Little et al., 1997; Ufkes and Kroon, 2012). The strength of the upwelling episodes, as well as their seaward expansion, are linked to wind stress (Lutjeharms and Meeuwis, 1987; Ufkes and Kroon, 2012) and glacial-interglacial fluctuations (Rosell-Melé et al., 2014). The

BUS comprises northern, central, and southern areas which differ from the duration of the upwelling and the level of productivity (Lutjeharms and Stockton, 1987; Ufkes et al., 2000; Petrick et al., 2015). Specifically, the northern and central areas (north of 30°S) are characterized by perennial upwelling and high productivity (Andrews and Hutchings, 1980; Lutjeharms and Stockton, 1987; Hutchings et al., 2009), with upwelled cold and high-nutrient waters extending offshore in filaments (Rosell-Melé et al., 2014). Conversely, the southern area (south of 30°S) experiences periodical (seasonal) upwelling

events and low nutrient levels (Andrews and Hutchings, 1980; Rosell-Melé et al., 2014; Petrick et al., 2015). The BUS is strictly linked to the Benguela Offshore Current (BOC) and the Benguela Coastal Current (BCC) (Fig. 2; Little et al., 1997; Rosell-Melé et al., 2014). The BOC is a cold surface water mass which flows along the African coast from Cape Town (34°S) to Walvis Bay (23°S). When reaching ~23°S, the BOC diverges to the west at the TGW, while its coastal branch (the BCC) continues to move northwards towards the Angola region (Little et al., 1997).

The boundaries of the BUS are represented in the south by the retroflection region of the Agulhas Current (AgC) and the Angola-Benguela Front (ABF) in the north (Fig. 2; Little et al., 1997), respectively. The AgC is a warm and saline current which flows along the eastern margin of Africa, bringing Indian Ocean waters to the South Atlantic Ocean (Bé and Duplessy, 1976; Olson and Evans, 1986; Fine et al., 1988; Petrick et al., 2015). Specifically, the AgC does not reach the southeast Atlantic region, but retroflects near the Cape Basin due to the Westerlies stress curl (Lutjeharms, 1981; Petrick et al., 2015). AgC

retroflection eddies (rings) of warmer and saline waters then leak into the BUS, moving northwest (Petrick et al., 2015). This influx of AgC waters from the Indian Ocean is known as Agulhas leakage (Fine et al., 1988; Petrick et al., 2015; Friesenhagen, 2022). A greater input of the Indian Ocean subtropical waters is induced by the increase in strength of the BOC (Garzoli et al., 1996). Conversely, the northward fluctuations of the cold Antarctic Circumpolar Current (ACC) result in a weaker leakage (McClymont et al., 2005; McIntyre et al., 1989; Peeters et al., 2004).

North of the BUS, the tropical warm and oligotrophic Angola Current (AC) flows poleward down to about 16°S, where it encounters the cool and nutrient-rich waters of the Benguela system, producing a thermal front known as the Angola-Benguela Front (ABF) (Fig. 2; Mohrholz et al., 2008; Kopte et al., 2017). The northern sector of the ABF is occupied by low-oxygen and high-saline subsurface waters (100-500 m water depth) named South Atlantic Central Water (SACW; Mohrholz et al., 2008; Ufkes and Kroon, 2012).





## 2 Materials and Methods

The studied sites (U1575, U1576) were cored during IODP Expedition 391, along the TGW Ridge seamount track, following a NE-SW transect. Lithological descriptions of the investigated sites mentioned below are from Sager et al. (2022). Sample depths are indicated as drilling depth below the seafloor (CSF-B, specified as mbsf) to avoid core overlaps originating from core expansion on deck, resulting in a recovery of more than 100% (as occurred in Hole U1576A).

Hole U1575A (21°51.9659′S, 06°35.4369′E; 3231.3 m water depth) is situated on the northwestern side of the Walvis Ridge, between Frio Ridge and Valdivia Bank (Fig. 1), and near the Namibian continental margin. Moreover, the site is located in the northernmost sector of the BUS (~17-25°S), under the influence of the BOC. The Pleistocene succession at this hole is represented by unconsolidated calcareous nannofossil-foraminiferal oozes (Lithostratigraphic Unit I; 0–~40 mbsf; Fig. 3). Gray-white and green bandings were recorded within the unit due to the accumulation of pyrite framboids and Fe-Mn-rich particles.

Hole U1576A (24°35.7520′S, 05°7.3163′E) was retrieved on the western side of the Valdivia Bank volcanic edifice at a water depth of 3032.3 m (Fig. 1). As for Site U1575, this hole is situated in the northern area of the BUS, within the latitudinal band of the BOC. The recovered Pleistocene sequence consists of part of Unit I (0-~45 mbsf) separated in subunits IA and IB, based on different sediment colors and subtle changes in clay content. Subunit IA is composed of pale brown nannofossil-foraminiferal oozes, while subunit IB is dominated by white nannofossil-foraminiferal oozes (Fig. 3).

### 2.1 Samples preparation

A total of 53 (Hole U1575A) and 41 (Hole U1576A) samples were prepared for biostratigraphic and quantitative analyses of the planktonic foraminiferal assemblage, respectively. The sediment was dried overnight at 40°C, using an electric oven. Subsequently, the dried sediment was soaked in distilled water and then wet-sieved into four-size fractions (63-125 µm, 125-250 µm, 250-500 µm and greater than 500 µm). In order to prevent contamination, all sieves were cleaned with methylene blue to recognize foraminiferal specimens from a previous wash. Once dried at 40-50°C, the sediment was transferred into labeled glass vials (Haynes, 1981; Snyder and Huber, 1996; Arrigoni et al., 2023). Thereafter, planktonic specimens were observed using a ZEISS DISCOVERY.V8 stereomicroscope and picked in the 125-500 µm size fractions. ZEISS DSM 982 (Gemini) scanning electron microscope (SEM) was used to better assess the state of preservation of planktonic foraminifera as well as to image the most relevant taxa (see Appendices A-B).

Calcareous nannofossils were analyzed in 61 samples and solely used for biostratigraphic investigations at the studied sites. The simple smear slide technique (e.g., Haq and Lohmann, 1976, Backman and Shackleton, 1983, Bown and Young, 1998) was performed to obtain microslides. Untreated sediment and a few drops of distilled water were mixed to create a suspension which was subsequently smeared on a coverslip with the use of a toothpick. The cover slip was then dried at ~50°C on a hotplate. Norland optical adhesive was utilized to mount the microslides. After preparation, slides were scanned for calcareous





nannofossil content with a standard light microscope Zeiss Lab.A1 model Axio at 1000 x magnification. Thereafter, individuals were imaged using light microscope Axioplan2 and camera Leica DFC 320 (see Appendices A-B). SEM analysis was

additionally performed to detect the presence of small-size coccoliths (e.g., *Emiliania huxleyi*).

## 2.2 Taxonimic remarks

### 2.2.1 Planktonic foraminifera

The taxonomy of planktonic foraminifera largely derived from Blow (1969), Postuma (1971), Rögl (1974), Kennett and Srinivasan (1983), Bolli and Saunders (1985), Chaisson and Leckie (1993), Loeblich and Tappan (1994), Weiner et al. (2015), Wade et al. (2018), and Bylinskaya (2004).

In this study, we differentiated *Neogloboquadrina pachyderma* from *Neogloboquadrina incompta* because they are considered two distinct species based on biogeographic, ecological and genetic differentiation (Brummer and Kroon, 1988; Darling et al., 2006). Specifically, we assigned the right coiling type to *Neogloboquadrina incompta* while the left coiling type to *Neogloboquadrina pachyderma* (Darling et al., 2006). Both *Neogloboquadrina pachyderma* and *Neogloboquadrina incompta* were separated from *Neogloboquadrina dutertrei* following Lam and Leckie (2020). The species name *Neogloboquadrina*

*dutertrei* was given to individuals possessing 5-6 chambers and with an open and deep umbilical area. Conversely, *Neogloboquadrina pachyderma* and *Neogloboquadrina incompta* show 4 to 4.5 chambers in the final whorl, a subquadrate to quadrate outline and a less open and deep umbilicus. Furthermore, we identified as *Neogloboquadrina acostaensis* all the specimens showing 5-5.5 chambers in the final whorl with straight sutures on both umbilical and spiral sides, and a wide apertural rim/plate (Lam and Leckie, 2020). This species differs from *Neogloboquadrina dutertrei* by showing a narrower

umbilicus with the rim/plate covering most of the umbilical area (Kennett and Srinivasan, 1983).

### 2.2.2 Calcareous nannofossils

The identification of calcareous nannofossils was based on "The Handbook of Cenozoic Calcareous Nannoplankton" Volumes 1–4 (Aubry, 1984, 1988, 1989, 1990), Perch-Nielsen (1985a, 1985b), Young (1998), Wade and Bown (2006), Bown and

Dunkley Jones (2012), and Nannotax3 (Young and Bown, 2017).

The taxonomy of *Reticulofenestra asanoi* follows Maiorano and Marino (2004). Specifically, circular to subcircular specimens without slits and larger than 6 µm were identified as *Reticulofenestra asanoi*. Conversely, subcircular morphotypes ≥ 5 µm, with (few) slits and a wider central area, were indicated as *Reticulofenestra* sp. All elliptical reticulofenestrids larger than 5 µm and with a central opening, were classified as *Reticulofenestra pseudoumbilicus* (see Young, 1990).



*Gephyrocapsa* placoliths were differentiated based on the size ranges defined by Raffi (2002): small (<4 µm), medium (4-5.5 µm), and large (>5.5 µm). The differentiation between *Calcidiscus tropicus* and *Calcidiscus macintyrei* follows Young (1998): individuals greater than 10 µm were assigned to *Calcidiscus macintyrei* and those smaller than 10 µm to *Calcidiscus tropicus*.

**2.3 Sample preservation**

**2.3.1 Planktonic foraminifera**

The preservation of planktonic foraminifera was rated as follows (see Tables S1-S2):
VG = very good (shells exhibiting an absence of recrystallization and overgrowth with all specimens recognizable at the
species level; G = good (tests showing only minor signs of recrystallization and overgrowth, with almost all the individuals identified at species level); M = moderate (common recrystallization and overgrowth observed on the foraminiferal shells and most individuals recognizable at the species level; P = poor (shells exhibiting substantial recrystallization and overgrowth, with identification at the species level often very difficult).

**2.3.2 Calcareous nannofossils**

Preservation for calcareous nannofossils was evaluated as follows (see Tables S1-S2):
VG = very good (specimens do not exhibit dissolution and overgrowth, with all the diagnostic features preserved); G = good (specimens show minor dissolution and overgrowth, with morphological characteristics slightly altered); M = moderate (individuals exhibit moderate dissolution, overgrowth and etching. Not all specimens were recognizable at the species level);
P = poor (specimens show high grade of dissolution, overgrowth, and etching. Morphological features are highly affected and most of the individuals were not identifiable at the species level).

**2.4 Biostratigraphy**

Biostratigraphic events for planktonic foraminifera were obtained from Gradstein et al. (2020) whereas age assignments for calcareous nannofossils were based on Wei (1993), Raffi (2002), and Gradstein et al. (2020).
The biozonation for calcareous nannofossils follows Backman et al. (2012), while planktonic foraminifera biostratigraphic zones are according to Wade et al. (2011). Bioevents were defined using base (B) and top (T) as well as the base common (Bc) and top common (Tc) occurrences of marker taxa (see Tables S1-S2 and Tables 1 and 2).

**2.5 Statistical analyses and ordination**



At least 300 individuals per sample were picked and identified for the paleoecological investigation of the planktonic foraminifera assemblages. The relative abundance of the recognized species was expressed as a percentage of the total count

of individuals in each sample (Tables S3-S4).

Statistical and ordination analyses in this study include the Similarity percentage analysis (SIMPER), cluster analysis (UPGMA) and principal component analysis (PCA), which were executed using the software PAST (version 4.09) (Hammer et al., 2001). Planktonic foraminifera relative abundances were arcsine root transformed (e.g., Sokal and Rohlf, 1995) before the multivariate analyses to ensure a normal distribution of the data values (e.g., Auer et al., 2019; Del Gaudio et al., 2023).

Using the Bray-Curtis similarity index, the Unweighted Pair Group Method with Arithmetic mean (UPGMA), was computed to define the clusters (Tables S3-S4 and Fig. 4). Moreover, the Bootstrapping (N=1000) function was used to test the stability of the clusters. The contribution of the species to the clusters was evaluated using the SIMPER analysis (using Bray-Curtis similarity). The clustering was performed, including and excluding biostratigraphic markers from the dataset, to assess their influence on the distributions of the clusters.

PCA was also performed to detect the principal environmental components (variables) controlling the planktonic foraminifera assemblages as well as to assess the results obtained from the cluster analysis (Figs. S5-S6). Taxa with an average percentage lower than 2% were excluded from statistical and ordination analyses. Species belonging to the *Trilobatus* plexus were grouped together as they genetically represent a single biological taxon (Hemleben et al., 1987; André et al., 2013). Specimens belonging to the tropical/subtropical morphotypes *Globigerinoides ruber* sensu strictu (s.s) and sensu latu (s.l.) (see Wang,

2000; Jayan et al., 2021; Del Gaudio et. al., 2023) were also lumped together as *Globigerinoides ruber* (white). This is because their abundances are too low to possibly infer any valuable variations in the paleoecological conditions at middle latitudes.

The *Globorotalia truncatulinoides* coiling ratio was computed as it reflects paleoenvironmental variations, such as the thermal structure of the upper water column (e.g., Thiede, 1971; Pickard and Emery, 1991). Specifically, the ratio was obtained using the total test count of the left and right morphotypes, normalized to 0 (Tables S3-S4).

The Agulhas leakage represents an inflow of warm and saline waters from the Indian Ocean to the southeastern Atlantic Ocean (Fine et al., 1988; Petrick et al., 2015; Friesenhagen, 2022). To evaluate the capability of the possible water exchanges between the Indian and the Atlantic Ocean (Tables S3-S4 and Fig. 5), we calculated the Agulhas Leakage Efficiency (ALE) Index (Caley et al., 2014). The index is expressed as follows:

$$ALE\ (\%) = (IOTG/(IOTG+SOG))*100$$

The Indian Ocean Tropical Group (IOTG) represents the species which strictly characterize the Agulhas water masses (e.g., *Globorotalia menardii*, *Trilobatus* spp.) while the Southern Ocean Group (SOG) is composed of taxa which thrive in cold, transitional, and subpolar water masses (e.g., *Globigerina bulloides*, *Globoconella inflata*, *Neogloboquadrina pachyderma*). Both IOTG and SOG are defined as the sum of the total count of each species belonging to the groups. We excluded all taxa with low abundances (<0.5%) or that are not indicative of either of the two groups, following Caley et al. (2014).




## 3 Results

### 3.1 Integrated Biostratigraphy

Integrated calcareous nannofossils' and planktonic foraminiferal biostratigraphic investigations allowed for the determination of 12 bioevents for Hole U1575A and 9 bioevents for Hole U1576A.

Detected bioevents, taxa and semi-quantitative abundance data for calcareous nannofossils and planktonic foraminifera are summarized in Tables 1-2 and Tables S1-S2. Biostratigraphic events are examined in detail in the discussion section.


### 3.2 Preservation and reworking

The preservation of planktonic foraminifera and calcareous nannofossils are indicated in Tables S1-S2. Planktonic foraminiferal tests were generally very well to well preserved in Holes U1575A and U1576A. However, several samples in

Hole U1575A (18.98-20.52 mbsf) showed good to moderate preservation, with specimens slightly affected by overgrowth and etching.

Calcareous nannofossils' preservation in Hole U1576A varies from good to very good, with most of the individuals exhibiting minor evidence of dissolution/overgrowth as well as displaying all their diagnostic features perfectly recognizable. Sediments in Hole U1575A contained well preserved specimens, with only two samples (between 18.98 and 20.52 mbsf) moderately

affected by dissolution/overgrowth.

Reworking of planktonic foraminifera and calcareous nannofossils was observed within the studied stratigraphic sequence at both sites (see Tables S1-S2). Its evaluation relied on the biostratigraphic distribution of the identified species, as well as changes in color and preservational state of the reworked forms compared to the in-situ assemblage. Planktonic foraminiferal Miocene to Early Pleistocene reworked taxa include *Globoconella miozea*, *Globoconella puncticulata*, and *Globigerinoides*

*bollii*. In Hole U1575A, the reworking mainly affected sediments between 18.27 and 19.02 mbsf in which a high number of reworked species was detected (see Tables S1-S2). Calcareous nannofossils instead showed Cretaceous, Paleogene to Miocene reworked forms such as *Bomolithus* spp., *Ericsonia* spp., *Helicosphaera vedderi*, and Discoaster *druggii*.

### 3.3 Planktonic foraminiferal assemblage distribution


A total of 6996 and 5540 planktonic foraminiferal specimens were identified for the assemblage study in Holes U1575A and U1576A, respectively. At both locations, the microfossil assemblages were dominated by planktonic foraminifera, whereas individuals of benthic foraminifera and ostracod shells were only sporadically observed. Planktonic foraminiferal relative abundance data and statistical results are shown in Tables S3-S4 and Figs. 3-5.





**Fig. 3:** Relative abundances (%) plotted against depth (mbsl) of the planktonic foraminiferal species which define the clusters in Holes U1575A and U1576A. The figure also shows the ratio between right and left-coiling tests of *Globorotalia truncatulinoides*. Foraminifera and calcareous nannofossils biozonations as well as the lithological units are also indicated.





### 3.3.1 Cluster analyses and ordination in Hole U1575A

UPGMA cluster analysis classified the samples into three main clusters (Table S3 and Fig. 4). The differentiation between the clusters was attained at a cut-off distance of ~0.81. The cophenetic correlation coefficient obtained from the application of UPGMA clustering is 0.7237. Additionally, cluster 1 can be separated into two subclusters (1a and 1b) with a cut-off score of ~0.83.

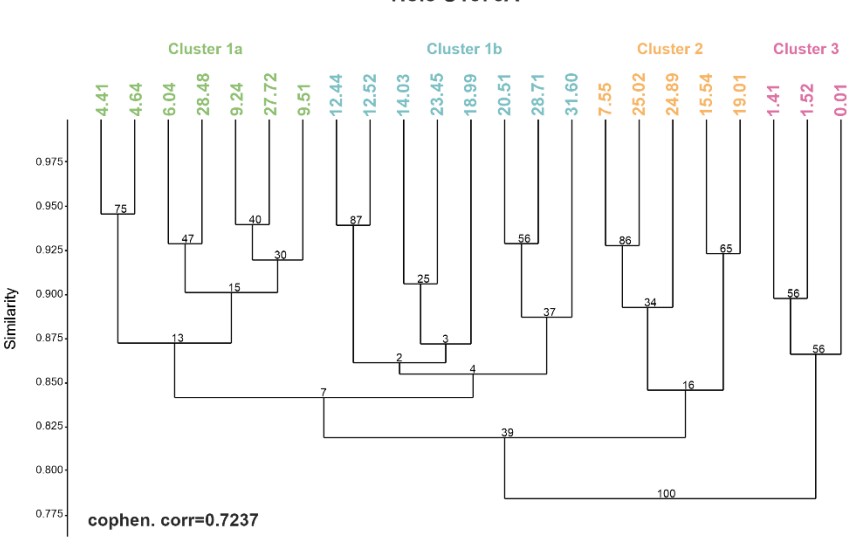

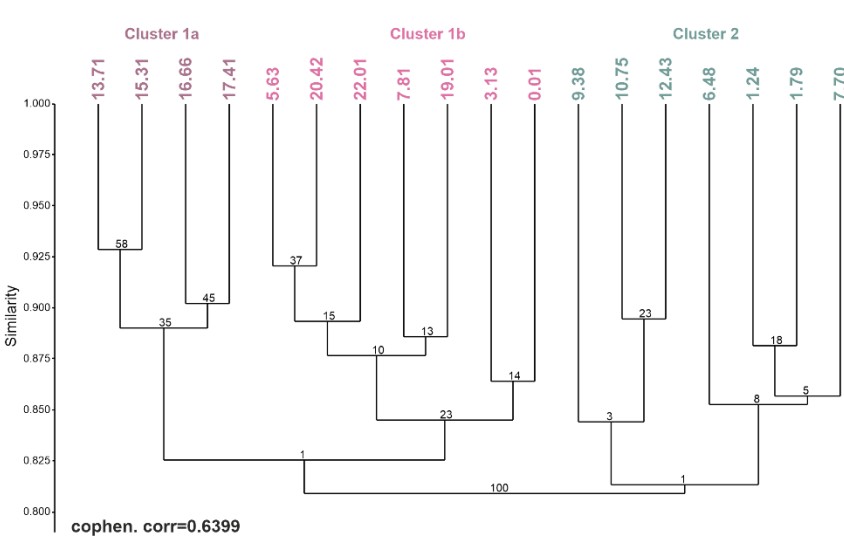

**Fig. 4:** UPGMA dendrograms of cluser analysis for Holes U1575A and U1576A.



SIMPER analysis (%) was used to identify which species mostly contributed to the cluster separation (Table S3 and Fig. 3).
The separation of clusters 1 and 2 largely relies on *Globoconella inflata* (contribution > 19%), *Globorotalia truncatulinoides*
sinistral and dextral forms, which account for 14.76% and 13.02% of the dissimilarity, respectively. Other species include
*Globorotalia crassaformis* (10.5%), *Globorotalia menardii* (9.59%), *Orbulina universa* (6.21%), and *Globigerina bulloides*
(5.7%).

Clusters 1 and 3 are distinguished based on *Globorotalia menardii* (27.05%), *Globorotalia crassaformis* (17.05%), and
*Globorotalia truncatulinoides* dextral (12.10%) and sinistral (8.33%). Minor contributors include *Neogloboquadrina*
*pachyderma* (7.30%), *Orbulina universa* (6.44%), and *Neogloboquadrina incompta* (5.85%). The abundance of *Globorotalia*
*menardii* (22.17%) and *Globoconella inflata* (18.18%) is the primary distinction between clusters 2 and 3. Other species
comprise *Globorotalia truncatulinoides* dextral (11.44%), *Globorotalia crassaformis* (8.59%), *Neogloboquadrina incompta*
(7.83%), and *Globorotalia truncatulinoides* sinistral (7.69%). The difference between subclusters 1a and 1b primarily depends
on *Globorotalia truncatulinoides* dextral (21.48%), *Globorotalia menardii* (13.06%), and *Globorotalia truncatulinoides*
sinistral (12.39%). Minor contributions derive from *Globoconella inflata* (9.48%), *Globigerinoides ruber* (7.48%), and
*Globorotalia crassaformis* (7.19%).

The average abundance (%) of planktonic foraminiferal taxa for each cluster/subcluster was also calculated and shown in Table
S3. Moreover, the abundances of the dominant species are plotted in Fig. 3. Planktonic foraminifera assemblage of subcluster
1a is dominated by *Globoconella inflata* (20.83%) and *Globorotalia crassaformis* (18.37%). Other common species include
*Neogloboquadrina incompta* (14.91%), *Globigerina bulloides* (7.08%), *Globorotalia truncatulinoides* dextral (6.37%) and
sinistral (6.01%). Foraminiferal association based on subcluster 1b showed a higher abundance of *Globoconella inflata*
(23.68%), *Globorotalia truncatulinoides* sinistral (8.52%) as well as lower values of *Globorotalia crassaformis* (14.26%) and
*Globorotalia truncatulinoides* dextral (0.58%), compared to subcluster 1a. Furthermore, the assemblage also comprises
*Neogloboquadrina incompta* (11.28%) and *Globigerina bulloides* (8.02%). The most representative taxa of cluster 2 are
*Globoconella inflata* (with the highest values of 40.08%), *Neogloboquadrina incompta* (15.52%), *Globorotalia crassaformis*
(9.50%), and *Globigerina bulloides* (7.18%).

The foraminiferal assemblage of cluster 3 shows consistently high abundances of *Globorotalia menardii* (19.91%) and
*Globoconella inflata* (19.19%) with low abundances of *Globorotalia crassaformis* (3.99%). This cluster is also characterized
by an increase in abundance of the tropical/subtropical taxa such as *Trilobatus* spp. (1.68%) and *Globoturborotalita rubescens*
335   (1.69%).

PCA analysis was performed to assess which species contribute the most to each principal component (Fig. S5). PCA results
indicate that three variables account for 76% of the variance (PC1=37.38; PC2=22.11; PC3=16.44%). *Globorotalia menardii*
and *Globorotalia truncatulinoides* dextral mainly dominate the component PC1, whereas negative loadings of PC1 largely
depend on *Globorotalia crassaformis* and *Globorotalia truncatulinoides* sinistral. The second principal component (PC2) is
largely positively correlated to *Globorotalia crassaformis* and *Globorotalia truncatulinoides* sinistral and dextral, while the
negative loadings are heavily related to *Globoconella inflata*. Principal component three (PC3) is positively associated with





high scores of *Globorotalia crassaformis* and *Globorotalia truncatulinoides* dextral, whereas *Globorotalia menardii* and *Globorotalia truncatulinoides* sinistral dominate the negative loadings.

The ALE Index, calculated on our dataset (Table S3), showed high percentages (33.50%) in cluster 3, whereas lower values
were observed for the remaining clusters (cluster 1a=12.52%; cluster 1b=17.79%; cluster 2=11.69%). The ratio between dextral and sinistral variants of *Globorotalia truncatulinoides* is shown in Fig. 3 and Table S3. The highest positive ratios were recorded in clusters 2 (average 2.55) and 3 (2.13), whereas the most negative values were obtained for cluster 1b (-0.93). Cluster 1a also shows positive values but far lower than those recorded for clusters 2 and 3. No correspondence was observed between the trend in abundance of *Globorotalia menardii* and the variation of the *Globorotalia truncatulinoides* ratio, plotted
against depth (see Fig. 5).



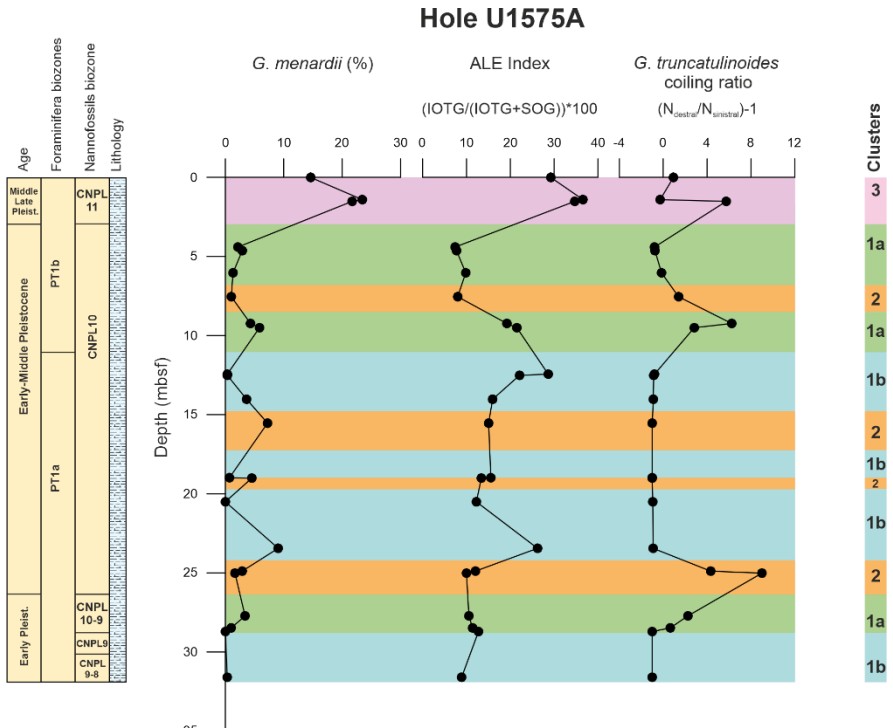

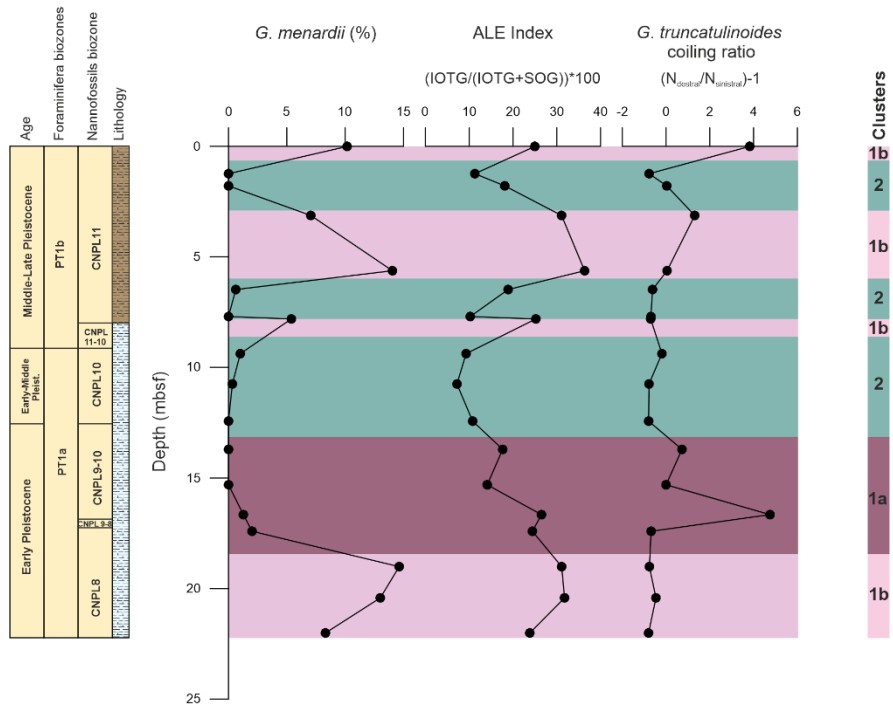

Gray/white nannofossil-foraminiferal oozes

Pale brown nannofossil-foraminiferal oozes



**Fig. 5:** The graphs represent relative abundances (%) of *Globorotalia menardii*, the ALE Index as well as the *Globorotalia truncatulinoides* dextral and sinistral coiling ratio. Foraminifera and calcareous nannofossil´ biozonations as well as the lithological units are also indicated.

### 3.3.2 Cluster analyses and ordination in Hole U1576A

Two main clusters were obtained in Hole U1576A using the UPGMA hierarchical clustering algorithm. Cluster 1 was separated into subclusters 1a and 1b (Table S4 and Fig. 4). The separation between the two main clusters was obtained with a cut-off distance of ~0.80.

Based on SIMPER analysis (Table S4), the difference between clusters 1 and 2 relies on *Globorotalia menardii* (contribution of 18.99%), *Globorotalia crassaformis* (17.19%), *Orbulina universa* (12.51%), and *Globorotalia truncatulinoides* sinistral

(12.27%). Other species comprise *Globorotalia truncatulinoides* dextral (8.21%) and *Neogloboquadrina incompta* (8.16%). Subclusters 1a and 1b are largely distinguished based on *Globorotalia menardii* (25.77%) and *Globorotalia crassaformis* (23.37%). Minor contributors include *Globorotalia truncatulinoides* dextral (9.87%), *Globorotalia truncatulinoides* sinistral (7.35%), and *Orbulina universa* (5.77%).

Average relative abundance data indicate that the most indicative species for subcluster 1a are *Globoconella crassaformis*

(17.43%) and *Globoconella inflata* (17.18%). Other species include *Globigerinoides ruber* (7.01%), *Globigerinita glutinata* (6.12%) and *Globigerina bulloides* (5.80%). The foraminiferal assemblage of subcluster 1b shows a high abundance of *Globoconella inflata* (17.49%), *Globorotalia menardii* (10.36%), and *Neogloboquadrina incompta* (9.90%). Other species comprise *Globigerina bulloides* (7.95%), *Globorotalia truncatulinoides* dextral (6.26%) and sinistral (6.43%), *Globigerinita glutinata* (5.05%), and *Globorotalia crassaformis* (4.52%). Cluster 2 is dominated by *Globoconella inflata* (20.79%),

*Globorotalia truncatulinoides* sinistral (12.56%), *Neogloboquadrina incompta* (13.16%), and *Orbulina universa* (11.12%), with minor contributions of *Globigerina bulloides* (9.05%) and *Globigerinoides ruber* (5.05%).

PCA analysis (Fig. S6) highlighted that three variables are responsible for 80.27% of the variance (PC1=31.90; PC2=28.71; PC3=19.66%). The first principal component (PC1) mainly depends on *Globorotalia crassaformis*. Conversely, PC1 is negatively correlated with *Globorotalia menardii* and *Globorotalia truncatulinoides* dextral and sinistral. *Globorotalia*

*menardii* positively dominates the PC2, while *Globorotalia truncatulinoides* sinistral, *Neogloboquadrina incompta*, and *Globoconella inflata* are negatively correlated to PC2. PC3 positive loadings mainly rely on *Neogloboquadrina incompta* and *Globorotalia truncatulinoides* sinistral. In contrast, *Orbulina universa* dominates the negative loading with a minor contribution of *Globorotalia menardii* and the right coiling type of *Globorotalia truncatulinoides*.

The ALE Index for Hole U1576A exhibits the highest percentage (average 29.19%) in cluster 1b, whereas clusters 1a and 2

show lower values (20.67 and 12.24 %, respectively; see Table S4).

The *Globorotalia truncatulinoides* dextral/sinistral ratio (see Table S4 and Fig. 3) shows positive values for cluster 1a (average 1.19) and 1b (0.35). Cluster 2 exhibits negative values of the ratio (-0.545). As also previously observed for Hole U1575A, the relative abundances (against depth) of *Globorotalia menardii* do not follow the changes in abundances of the right and left variants of *Globorotalia truncatulinoides* (see Fig. 5).



## 4 Discussion

### 4.1 Integrated Biostratigraphy

Integrated calcareous nannofossils and planktonic foraminifera biostratigraphy enabled to provide a well-defined stratigraphic
record for Holes U1575A and U1576A, as well as to improve the biostratigraphic resolution obtained during the IODP
shipboard investigations (Sager et al., 2023).

#### 4.1.1 Hole U1575A

The studied stratigraphic sequence spans from Late Pliocene (Piacenzian) to Quaternary according to planktonic foraminiferal
and calcareous nannofossil datums (Table 1; Table S1). The oldest detected bioevent at Hole U1575A is the top occurrence
(T) of *Dentoglobigerina altispira* (Sample U1575A-5R-7W, 0-2 cm; 46.89 mbsf), which was recorded in the Atlantic Ocean
at 3.13 Ma (Wade et al., 2011; Gradstein et al., 2020). The abovementioned sample also contains the base (B) of the planktonic
taxon *Globoconella inflata* (B 3.24 Ma; Gradstein et al., 2020). The concomitant presence of *Dentoglobigerina altispira* and
*Globoconella inflata* allowed the assignment of the sample to planktonic foraminifera zones PL3-PL4 (Wade et al., 2011) and
calcareous nannofossil zone CNPL4 (Backman et al., 2012). Sediments between 46.89 and 44.20 mbsf were dated younger
than 3.13 and older than 1.98 Ma (PL4-PL6 and CNPL4-CNPL6 zones), based on the absence of *Dentoglobigerina altispira*
and *Globorotalia truncatulinoides* (B 1.93 Ma; Gradstein et al., 2020).

| Bioevent | Age (Ma) | Sample ID | Depth (mbsf) | Reference |
|---|---|---|---|---|
| B *G. calida* | 0.22 | 1R-1W, 0-2 | 0.02 | Gradstein et al. 2020 |
| B *E. huxleyi* | 0.29 | 1R-1W, 140-142 | 1.42 | Gradstein et al. 2020 |
| B *G. flexuosa* | 0.40 | 1R-2W, 0-2 | 1.53 | Gradstein et al. 2020 |
| T *P. lacunosa* | 0.43 | 1R-3W, 138-140 | 4.40 | Gradstein et al. 2020 |
| T *G. tosaensis* | 0.61 | 2R-2W,143-145 | 12.43 | Gradstein et al. 2020 |
| B *G. hessi* | 0.74 | 3R-1W, 0-2 | 19.02 | Gradstein et al. 2020 |
| T common *R. asanoi* | 0.91 | 3R-6W, 0-2 | 26.51 | Gradstein et al. 2020 |
| B common *R. asanoi* | 1.14 | 3R-7W, 65-67 | 28.49 | Gradstein et al. 2020 |
| T *H. sellii* | 1.24 | 4R-2W, 138-140 | 31.59 | Gradstein et al. 2020 |
| T *N. acostaensis* | 1.58 | 4R-3W, 88-90 | 32.59 | Gradstein et al. 2020 |
| B *G. truncatulinoides* | 1.93 | 5R-4W, 137-139 | 44.20 | Gradstein et al. 2020 |
| T *D. altispira* | 3.13 | 5R-7W, 0-2 | 46.89 | Gradstein et al. 2020 |

**Table 1:** Calcareous nannofossil and planktonic foraminifera bioevents detected in Hole U1575A. B=base; T=top; Bc=base common
occurence; Tc=top common occurrence.

The top occurrence of *Neogloboquadrina acostaensis* (1.58 Ma; Gradstein et al., 2020) occurred at 32.59 mbsf (Sample
U1575A-4R-3W, 88-90 cm). The sediment between the basal occurrence of *Globorotalia truncatulinoides*, and the top



occurrence of *Neogloboquadrina acostaensis* encompasses the biozones PL6-PT1a and CNPL6-CNPL8. Among calcareous nannofossils, *Calcidiscus macintyrei* (T 1.60 Ma; Gradstein et al., 2020) and *Gephyrocapsa* spp. > 5.5 μm (B 1.59 Ma;

Gradstein et al., 2020) were also detected in the abovementioned depth interval. However, *Gephyrocapsa* spp. > 5.5 μm was present only in two samples (U1575A-4R-2W, 138-140 cm and U1575A-4R-3W, 88-90 cm; 31.59-32.61 mbsf), whereas the stratigraphic appearance of *Calcidiscus macintyrei* was rare and scattered, making the placement of the bioevent extremely difficult. Furthermore, its last occurrence is considered poorly accurate due to ambiguous taxonomic identifications (Raffi et al., 1995; Raffi et al., 2006). For all the reasons discussed above, the use of *Neogloboquadrina acostaensis* as a biostratigraphic

marker is preferred here.

The top occurrence of *Helicosphaera sellii* (1.24 Ma; Gradstein et al., 2020) is located at 31.59 mbsf (Sample U1574A-4R-2W, 138-140 cm). This bioevent is well-defined (Raffi et al., 2006) and is considered isochronous in the equatorial and mid-latitude sectors of the Atlantic Ocean (Gradstein et al., 2020). The stratigraphic interval between the top occurrences of *Neogloboquadrina acostaensis* and *Helicosphaera sellii* falls within zones PT1a and CNPL8-CNPL9.

Two important biostratigraphic horizons were detected between 28.49 and 26.51 mbsf: the base common (Bc) and the top common (Tc) occurrences of *Reticulofenestra asanoi* (1.14 and 0.91 Ma, respectively; Gradstein et al., 2020) were used to restrict the depth interval to zones PT1a and CNPL9-CNPL10. The stratigraphic distribution of *Reticulofenestra asanoi* is constrained to Early-Late Pleistocene (Sato et al., 1991; Wei, 1993; Raffi, 2002; Maiorano and Marino, 2004), with the first and last common occurrences of the species considered as more reliable than its absolute first and last appearances (Maiorano

and Marino, 2004).

Sediments above the top common *Reticulofenestra asanoi* and the basal occurrence of the *Globorotalia hessi* (0.74 Ma; Gradstein et al., 2020) fall within zones PT1a and CNPL10. Calcareous nannofossil assemblages in this depth interval (19.02-26.51 mbsf) also contain a few specimens of *Reticulofenestra asanoi* and *Reticulofenestra* sp. However, while the occurrences of *Reticulofenestra asanoi* are discontinuous above its Tc (Maiorano and Marino, 2004), the placement of the last appearance

of *Reticulofenestra* sp. was found to be inconsistent in previously studied Atlantic sections (Maiorano and Marino, 2004). Thus, the use of B *Globorotalia hessi* as bioevent is favored here.

*Globorotalia tosaensis* shows a fairly continuous stratigraphic distribution in Hole U1575A (between 44.20 and 12.43 mbsf), and its top occurrence (0.61 Ma; Gradstein et al., 2020) was used to determine the base of subzones PT1b (Wade et al., 2011) which, in turn, corresponds to zone CNPL10 of Backman et al. (2012). The concomitant extinctions of *Globorotalia ronda* at

~0.6 Ma (Bylinskaya, 2005; Aze et al., 2011) in the stratigraphic sequence further support the validity of the discussed bioevent. The top occurrence of *Pseudoemiliania lacunosa* (0.43 Ma; Gradstein et al., 2020) and basal occurrence of *Globorotalia flexuosa* (0.40 Ma; Gradstein et al., 2020) occurred at 4.40 and 1.53 mbsf, allowing to assign the sediments to zones PT1b and CNPL10-CNPL11.

The topmost biostratigraphic events detected in Hole U1575A were the base of *Emiliania huxleyi* (0.29 Ma; Gradstein et al.,

2020) and *Globigerinella calida* (0.22 Ma; Gradstein et al., 2020), detected at 1.42 and 0.02 mbsf, correspondingly. The previously mentioned bioevents constrain the stratigraphic interval to zone PT1b and CNPL11.





### 4.1.2 Hole U1576A

The analyzed sediment interval in Hole U1576A comprises Early to Late Quaternary deposits (Table 2; Table S2).

The bottommost part of the stratigraphic sequence (28.58 mbsf) is dated older than 1.98 Ma but younger than 3.24 Ma based on the top occurrence of *Globigerinoides extremus* (Gradstein et al., 2020) and the basal appearance of *Globoconella inflata* (B 3.24 Ma; Gradstein et al., 2020). The assigned biozones for the interval were PL6-PL3 and CNPL6-CNPL4 for planktonic foraminifera and calcareous nannofossils, respectively.

| Bioevent | Age (Ma) | Sample ID | Depth (mbsf) | Reference |
|---|---|---|---|---|
| B *G. calida* | 0.22 | 1R-3W, 60-62 | 3.14 | Gradstein et al. 2020 |
| B *G. flexuosa* | 0.40 | 2R-1W, 0-2 | 7.82 | Gradstein et al. 2020 |
| T *G. tosaensis* | 0.61 | 2R-2W, 10-12 | 9.37 | Gradstein et al. 2020 |
| B *G. hessi* | 0.74 | 2R-4W, 20-22 | 12.44 | Gradstein et al. 2020 |
| B *R. asanoi* | 1.17 | 2R-7W, 0-2 | 16.67 | Wei 1993; Raffi 2002 |
| T *G. obliquus* | 1.30 | 3R-1W, 0-2 | 17.40 | Gradstein et al. 2020 |
| T *N. acostaensis* | 1.58 | 3R-5W, 0-2 | 23.40 | Gradstein et al. 2020 |
| T *D. broweri* | 1.93 | 4R-1W, 60-61 | 27.60 | Gradstein et al. 2020 |
| T *G. extremus* | 1.98 | 4R-2W, 10-12 | 28.58 | Gradstein et al. 2020 |

**Table 2:** Calcareous nannofossil and planktonic foraminifera bioevents detected in Hole U1576A. B=base; T=top; Bc=base common occurence; Tc=top common occurrence.

The top occurrence of *Discoaster broweri* (1.93 Ma; Gradstein et al., 2020) was recorded at 27.60 mbsf, constraining the age of the sediments to zones PL6 and CNPL6. The following biohorizon is represented by the top occurrence of

*Neogloboquadrina acostaensis* (1.58 Ma; Gradstein et al., 2020), which constrains the sediment between 27.60 and 23.40 mbsf to zones PL6-PT1a and CNPL6-CNPL8. Similarly to Hole U1575A, the occurrence of *Calcidiscus macintyrei* (T 1.60 Ma; Gradstein et al., 2020) was not considered at this site. The last appearance datum of *Globigerinoides obliquus* (T 1.30 Ma; Gradstein et al., 2020) was detected at 17.40 mbsf, constraining the interval below to zones PT1a and CNPL8.

The stratigraphic interval observed at Site U1575, characterized by a distinct increase in abundance of *Reticulofenestra asanoi*,

was not recorded at Hole U1576A. However, it was possible to use the first absolute occurrence of *Reticulofenestra asanoi* (B 1.17 Ma; Raffi, 2002) to assign the sediments between this biohorizon and T *Globigerinoides obliquus* to zones PT1a and CNPL8-CNPL9. Although not as distinct as the Bc *Reticulofenestra asanoi* event (1.14 Ma; Gradstein et al., 2020) the first appearance datum of *Reticulofenestra asanoi* is still considered as a useful biohorizon in Pleistocene deposits (Takayama and Sato, 1987; Wei, 1993; Raffi, 2002).

The base of *Globorotalia hessi* (0.74 Ma; Gradstein et al. 2020) was observed in Sample U1576-2R-4W, 20-22 cm (12.44 mbsf). This bioevent together with the B of *Reticulofenestra asanoi* constrained the age of the sediments to PT1a and CNPL9-



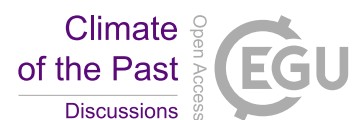

CNPL10. The top of the planktonic foraminiferal subzone PT1a was defined at Hole U1576A by the top occurrence of *Globorotalia tosaensis* (0.61 Ma; Wade et al., 2011). The sediments between B *Globorotalia hessi* and T *Globorotalia tosaensis* were assigned to zones PT1a and CNPL10.

The youngest biostratigraphic events observed at this site were the base of *Globorotalia flexuosa* (0.40 Ma; Gradstein et al., 2020) and *Globigerinella calida* (0.22 Ma; Gradstein et al., 2020), occurring at 7.82 and 3.14 mbsf, respectively. These bioevents allowed us to assign sediments to zones PT1b and CNPL11.

## 4.2 Paleoceanographic evolution in the northern Benguela system inferred from the planktonic foraminiferal
assemblages

Variations of the planktonic foraminiferal assemblages during the Quaternary can be interpreted as indicative of changes in the water mass dynamics within the northern Benguela system. In this respect, quantitative analyses performed on the foraminiferal communities (UPGMA and PCA) revealed different paleoceanographic settings in Holes U1575A and U1576A.

Specifically, UPGMA defined several clusters representing distinct planktonic assemblages reflecting different environmental conditions. Furthermore, the use of PCA allowed to define which paleoenvironmental variables affect the planktonic assemblages.

### 4.2.1 Hole U1575A

Three major clusters were defined for Hole U1575A, reflective of the existence of three main paleoceanographic conditions, as follows (Figs. 3-5 and 6; Table S3):

Normal BOC conditions (Cluster 2): This cluster is dominated by *Globoconella inflata*, representing 40% of the total assemblage. *Neogloboquadrina incompta* and *Globigerina bulloides* also show common abundances. The three abovementioned species are commonly found within the BOC, which represent the relatively oligotrophic and less cold (17-

22°C; Rouault and Tomety, 2022) offshore component of the Benguela current system (Giraudeau, 1993; Little et al., 1997; Ufkes and Kroon, 2012). Conversely, species like *Neogloboquadrina pachyderma* and *Turborotalita quinqueloba* (constantly exhibiting low abundances in our record) thrive in the more nutrient-rich and cooler (15-17°C; Rouault and Tomety, 2022) surface waters of the BCC and constitute the typical upwelling fauna (Giraudeau, 1993; Little et al., 1997). Specifically, the abundances of *Globoconella inflata*, *Neogloboquadrina incompta*, and *Globigerina bulloides* were previously found to

increase offshore (away from the coast) based on several retrieved cores (Giraudeau, 1993). *Globorotalia crassaformis* is typically associated with warm and low-oxygenated subsurface waters (SACW) situated in the Angola Basin, north of the ABF (van Leeuwen, 1989; Oberhänsli et al., 1992; Ukfes and Kroon, 2012). Thus, the variation in abundance of *Globorotalia crassaformis* reflects a north-south shifting of the thermal ABF (Ukfes and Kroon, 2012), with higher values indicating southward fluctuations of the ABF and the expansion of the Angola warm waters within the northern Benguela region


(Shannon et al., 1986; Monteiro and van der Plas, 2006). In cluster 2, this taxon exhibits low abundances, accounting for only 9.50% of the assemblage. Thus, we interpreted the foraminiferal association of cluster 2 as indicative of what we described as normal Benguela conditions. This definition refers to a system where the ABF is located north of the Benguela region so that the BOC waters are not perturbed by the southward intrusions of the SACW (Fig. 6). This is further corroborated by the high amount of *Globorotalia inflata*, which thrives in the cooler water of the BOC and shows an opposite trend in abundance

compared to *Globorotalia crassaformis*. The right and sinistral-coiling types of *Globorotalia truncatulinoides* show distinct environmental conditions, as underlined by several studies (e.g., Herman, 1972; Lohmann and Schweitzer, 1990; Billups et al., 2016) with the ratio between dextral and sinistral specimens used as a proxy for the water column structure (e.g., the depth of the thermocline) (Feldmeijer et al., 2014; Billups et al., 2016). *Globorotalia truncatulinoides* sinistral prefers warmer and less productive waters with a more stable and deeper thermocline (e.g., the center of the gyre system; Herman, 1972; Billups

et al., 2016). Conversely, *Globorotalia truncatulinoides* dextral is documented to prefer a shallower habitat in the water column and is associated with cooler and more productive waters (Feldmeijer et al., 2014; Billups et al., 2016). In cluster 2, *Globorotalia truncatulinoides* dextral and sinistral exhibit similar abundances (~6%), indicating the presence of paleoecological conditions which favor both coiling types. This is in agreement with the proposed paleoenvironmental model for cluster 2, as the persistence of normal BOC and the absence of water mixing with SASG waters resulted in the presence of

cold-water temperature and a relatively stable thermocline in the region. PCA analysis further supports the paleoceanographic conditions interpreted for cluster 2 (Fig. S5), with PC2 recording cold, relatively stable water column conditions and ABF in a more northern position, as suggested by the negative and positive loadings of *Globoconella inflata* and *Globorotalia crassaformis*, respectively.

Limited SACW intrusions (Cluster 1b): The increase in abundance of *Globorotalia crassaformis* (14.26%) and the concomitant decrease of *Globoconella inflata* (23.68%) points to a southward shifting of the ABF with consequently mixing between the warm SAGW and the colder BOC. Our data suggest that the intensity of water mixing was sufficient to promote a weak instability of the thermocline (Fig. 6). This interpretation is corroborated by a sharp increase in the abundance of *Globorotalia truncatulinoides* sinistral compared to the dextral, as indicated by relative abundance data and negative values of the

*Globorotalia truncatulinoides* coiling ratio. The dominance of the sinistral variant is due to the fact that the low amount of water mixing caused a small increase in the instability of the thermocline and induced a rise in water temperature.

Expanded SACW intrusions (Cluster 1a): The abundance of *Globorotalia crassaformis* and *Globoconella inflata* continues to increase (18.37%) and decrease (20.83%), respectively. This data indicates an expansion of the SACW within the Benguela

region which, in turn, leads to a stronger water mixing, producing a higher thermocline instability compared to cluster 1b (Fig. 6). *Globorotalia truncatulinoides* dextral and sinistral exhibit again similar abundances (3.14 and 1.48%). This is because the water mixing produces a strong instability of the thermocline (favoring the dextral form) but at the same time induces a higher increase in water temperature (preferred by the sinistral variant) than those observed for cluster 1b. PCA results indicate the




southern shifting of the ABF and the increase of water mixing (Table S5). PC3 is positively linked with *Globorotalia*

*crassaformis* and negatively related to *Globoconella inflata*, reflecting the southward movement of the ABF. Moreover, the
positive and negative loadings of *Globorotalia truncatulinoides* dextral and sinistral highlight the variations in the intensity of
the water column instability, which has a higher impact on the dextral type (Fig. 6).

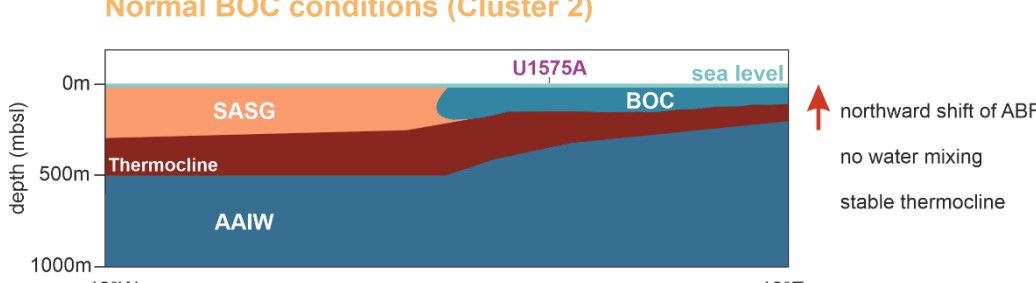

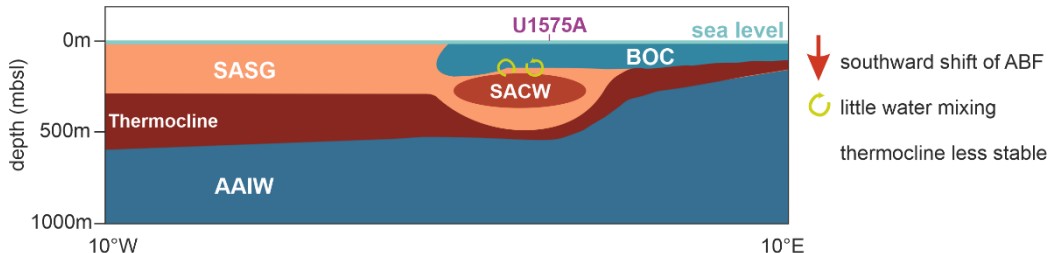

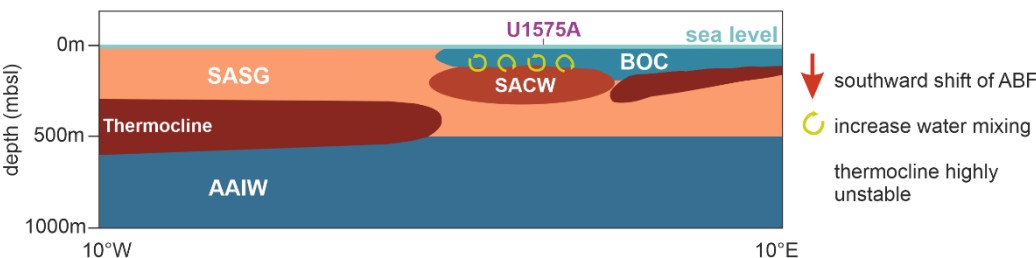

**Fig. 6:** Conceptual model showing the interaction between the SACW and BOC water masses in clusters 1 and 2 defined for Hole U1575A.
BOC=Benguela Offshore Current; SACW=South Atlantic Central Water; SASG=South Atlantic Subtropical Gyre waters; AAIW=Antarctic
Intermediate Water. The local thermocline is represented as a dark orange area.

Agulhas leakage within the Benguela system (Cluster 3): the planktonic foraminiferal assemblage shows an increase in tropical

taxa, among which *Globorotalia menardii* exhibits the highest abundances (~20%). Previous studies (Peeters et al., 2004;





Caley et al., 2012; Villar et al., 2015) hypothesized that the tropical fauna in the southeastern Atlantic represents a reseeding population from the Indian Ocean. Specifically, giant eddies of warm water from the Agulhas current can access the Benguela region according to a mechanism known as the Agulhas leakage (Fine et al., 1988; Petrick et al., 2015; Friesenhagen, 2022). The variation in the intensity of the Agulhas leakage through time allows the tropical fauna to overcome oceanographic barriers

(e.g., the STC; Fig. 2; Friesenhagen, 2022) and enter the Atlantic Ocean. Chaisson and Ravelo (1997) proposed an alternative scenario, asserting that changes in wind stress directions between the eastern and western sides of the Atlantic Ocean during the Pleistocene induced a deepening and a shoaling of the thermocline in the west and east Atlantic, respectively. This would promote more favorable conditions for *Globorotalia menardii* in the eastern region of the Atlantic gyre as this thermocline species responds to variation in the vertical water column stratification (Fairbanks et al., 1982; Curry et al., 1983; Friesenhagen,

2022). Our data indicate that the abundance trend of *Globorotalia menardii* does not consistently change with the *Globorotalia truncatulinoides* coiling ratio (Fig. 4), which, in turn, reflects variations of the thermocline in the water column (Feldmeijer et al., 2014; Billups et al., 2016). Moreover, PCA results show that *Globorotalia menardii* dominates the positive loading of PC1, with *Globorotalia crassaformis* showing negative scores (Table S5). Finally, the ALE Index (Fig. 5; Table S3) shows higher values (~33%) only in cluster 3 indicating an increase in Indian Ocean tropical fauna. Overall, our results assert that

*Globorotalia menardii* cannot be part of the SASG domain because, in this case, the variation in abundance of the taxon should respond to changes in the thermocline. *Globorotalia menardii* is also not a constituent of the Angola subtropical fauna as it is statistically not positively correlated to *Globorotalia crassaformis*. The increment of the Indian Ocean tropical taxa detected with the ALE Index lends further support to the hypothesis that the Agulhas leakage process is responsible for the reseeding of these tropical species in the southeast Atlantic realm.

The record of a subtropical Indian Ocean fauna in Hole U1575A indicates that the strength of the Benguela current must have been strong enough to allow the Agulhas eddies to reach the northernmost area of the BUS. In fact, a greater ingress of Agulhas waters within the BUS may be favored by an intense BOC and a more southern position of the subarctic front (Garzoli et al., 1996; McClymont et al., 2005; Peeters et al., 2004).

The input of large Agulhas eddies in the system can cause variations in the water column structure, affecting the thermocline

(Klein and Lapeyre, 2009). Specifically, in the southern hemisphere, anticyclonic (cyclonic) eddies induce a shoaling (deepening) of the thermocline. In the southeastern Atlantic Ocean, intermixing of anticyclonic warm eddies from the Agulhas with the cold Benguela current can produce instability of the water column and shoals the thermocline. This process is possibly recorded by PC1, which indicates positive loadings for *Globorotalia truncatulinoides* dextral and *Globorotalia menardii* and negative scores for *Globorotalia truncatulinoides* sinistral.


### 4.2.2 Hole U1576A

Cluster analysis for Hole U1576A resulted in two major clusters, with cluster 1 separated into subclusters 1a and 1b. The paleoenvironmental conditions associated with each cluster are discussed below (Fig. 3-4 and 5; Tables S4 and S6).





Expanded SACW intrusions (Cluster 1a): the foraminiferal assemblage of cluster 1a is characterized by high abundances of
*Globorotalia crassaformis* (17.43%), and slightly lower values of *Globoconella inflata* (17.18%) compared to clusters 1b and
2. The increase in abundance of *Globorotalia crassaformis*, with lower amounts of *Globoconella inflata*, indicates the
expansion of the SACW within the Benguela system, as also observed in Hole U1575A. However, the difference in abundance
between *Globorotalia crassaformis* and *Globoconella inflata* is not as marked as in Hole U1575A since Site U1576 is located
further offshore, where the effect of the BOC is less pronounced.

*Globorotalia truncatulinoides* dextral and sinistral show similar abundances (5.09% and 4.03%). We believe that the close
abundances of the dextral and sinistral variants are likely not linked to a strong instability of the thermocline, enhanced by the
intrusions of warm SACW (as observed for Hole U1575A). This is confirmed by PCA results which highlight a negative
correlation between *Globorotalia crassaformis* and both *Globorotalia truncatulinoides* dextral and sinistral (Fig. S6). A
possible explanation could be the peculiar position of Hole U1576A, which is located in a more southern position and closer
to the center of the gyre, compared to Hole U1575A. Specifically, it lies in an area where the more temperate distal portion of
the BOC encounters the warm waters of the subtropical gyre. This area is then characterized by relatively warmer waters
(favoring the sinistral variants) but also by the mixing between the BOC and the subtropical gyre waters (a condition preferred
by the dextral form of *Globorotalia truncatulinoides*). Waters entering from the Angola Basin (SACW) were probably already
mixed with the Benguela current before reaching the latitudinal position of Hole U1576A and cannot have a strong impact on
the thermocline stability (as instead observed for Hole U1575A).

Agulhas leakage within the Benguela system (Cluster 1b): Hole U1576A also recorded phases of increase in the Agulhas
leakage (Fig. 5; Table S4). The ALE Index shows higher values (~30%) for this cluster, corresponding to an increase in the
Indian Ocean tropical taxa within the assemblage. As observed for Hole U1575A, the abundance trend of *Globorotalia*
*menardii* does not follow the change in the ratio between *Globorotalia truncatulinoides* dextral and sinistral. Thus, the
variation in abundance of *Globorotalia menardii* is not linked to changes in the regional thermocline.
Similar to Hole U1575A, PCA results here also show positive loadings for *Globorotalia menardii* and *Globorotalia*
*truncatulinoides* dextral but a negative loading for *Globorotalia truncatulinoides* sinistral. This can be again explained by the
instability of the thermocline linked to the mixing between the Agulhas eddies with the Benguela waters. However, thermocline
variability was less pronounced than in Hole U1575A. This is indicated by the lower PCA loading values of *Globorotalia*
*truncatulinoides* dextral than those observed in Hole U1575A. We believe that the location of Hole U1576A (closer to the
center of the gyre) could again play an important factor in explaining the subtle variations of the thermocline for this site. In
fact, warm Agulhas eddies mix with already more temperate waters of the BOC, producing a smaller impact on the thermocline.

Nutrient filaments within the BOC (Cluster 2): this foraminiferal assemblage is dominated by *Globoconella inflata* with
common occurences of *Globorotalia truncatulinoides* sinistral (12.56%), *Neogloboquadrina incompta* (13.16%), and *Orbulina*
*universa* (11.12%). SIMPER analysis indicates that the main species responsible for the clustering are *Orbulina universa*,





*Globorotalia truncatulinoides* sinistral and dextral, and *Neogloboquadrina incompta*. Moreover, according to PCA results, the same species show the highest positive and negative loading scores for PC3.

The taxon *Orbulina universa* can inhabit tropical/subtropical as well as transitional water masses (Bé and Tolderlund, 1971; Schiebel and Hemleben, 2017) and can prefer waters with moderate nutrient level (van Leeuwen, 1989; Giraudeau, 1993; Lombard et al., 2011; Ufkes and Kroon, 2012). This species was commonly found in the northern Benguela region, south of the ABF (Bremner, 1983; Herbert, 1987), and restricted to a latitudinal range of about 17-25°S (Giraudeau, 1993). However, this study confirmed its presence until at least 24°S.

A previous study from Giraudeau (1993) indicated that the increase in abundance of *Orbulina universa* can be linked to the concomitant presence of warmer (>18°C) and more nutrient-rich conditions in the area. This paleoenvironmental interpretation is corroborated by our results showing a foraminiferal assemblage characterized by common *Orbulina universa* and species indicative of the less fertile offshore waters typical of the BOC (*Globorotalia inflata*, *Globigerina bulloides*, *Neogloboquadrina incompta*). Moreover, the presence of common *Globorotalia truncatulinoides* sinistral supports the warmer

water conditions (Fig. 5; Herman, 1972; Billups et al., 2016) at the site. PC3 (Fig. S6) indicates that the highest and opposite loadings are associated with *Neogloboquadrina incompta* (positive scores) and *Orbulina universa* (negative scores). *Neogloboquadrina incompta* thrives in the BOC and prefers temperatures between 10°C and 18°C. The co-existence of both species supports the presence of temperate and relatively nutrient-enriched waters. Interestingly, *Globorotalia truncatulinoides* dextral exhibits a negative score as this coiling-variant prefers more productive waters (Billups et al., 2016). However,

*Globorotalia truncatulinoides* sinistral shows a higher abundance because, although preferring less fertile waters, it thrives in warmer water masses away from the continental margin.

We believe that the assemblage of cluster 2 may reflect episodes of nutrient filaments transported offshore from the coastal upwelling zone. Ufkes et al. (2000) and Ufkes and Kroon (2012) detected phases of extreme coastal upwelling events in the northern BUS during the Pleistocene, possibly linked to powerful zonal winds. Those strong winds could transport a small part

of the nutrients, upwelled along the coast, further offshore within the more oligotrophic portion of the northern BUS.

### 4.3 Variations of the paleoceanographic conditions during the Quaternary

The planktonic foraminiferal records in Holes U1575A and U1576A indicate changes in the paleoenvironmental conditions

since the Early Pleistocene (Figs. 3,5 and 6). Specifically, we observed an alternation of periods reflective of variations in the ABF positions, Agulhas intrusions from the Indian Ocean and an increase of the nutrient transport offshore, further away from the coastal upwelling center.

The shifting of the ABF occurred several times within the whole studied record (Fig. 3). Particularly, its southward (northward) movements indicate the intrusion (absence) of the warm SACW within the northern sector of the BUS. Furthermore, the

southern extension of the ABF was accompanied by phases of limited/expanded SACW ingressions and their mixing with the more temperate waters of the BOC (Fig. 6). Several authors (Walter, 1937; Boyd and Thomas, 1984; Boyd et al., 1987;



Shannon and Nelson, 1996) linked the southern/northern shift of the ABF to the interannual Benguela Niño/Niña phenomena. During the Benguela Niño (Niña) events, an increase (decrease) in sea surface temperatures (SSTs) occurs in the eastern equatorial and south-east Atlantic Ocean due to the relaxation (intensification) of the trade winds (Rouault et al., 2007; Rosell-

Melé et al., 2014; Illig and Bachèlery, 2024). The reduction in intensity of the trade winds during the Benguela Niño induces warmer SST in the equatorial Atlantic and the ingress of warm Angola water in the BUS (Hisard, 1980; Philander, 1990; Illig et al., 2004). The opposite situation occurs during the Benguela Niña, when SSTs decrease and no SACW expands in the Benguela system. Thus, the Benguela Niño/Niña events have a strong impact on the upwelling intensity and water mixing (Boyer et al., 2001; Imbol Koungue and Brandt, 2021). Benguela Niño events are considered the Atlantic counterpart of the

widely-known Pacific El Niño-Southern Oscillation (ENSO; Bjerknes, 1969), which is linked to changes in the wind strength due to variations in the atmospheric circulation patterns in the Pacific Ocean (Qiu and Chen, 2010; Kaboth-Bahr and Mudelsee, 2022). However, it is important to note that no conclusive evidence exists that the Atlantic Benguela Niño can be in phase with the Pacific ENSO (Shannon and Nelson, 1996). In fact, they may reflect different forcing processes (Rosell-Melé et al., 2014). Modern SST data (Gammelsrød et al., 1998; Rouault et al., 2007) from the Angola and Cape basins indicate intervals of higher

sea-surface temperatures interpreted as the result of Benguela Niño events in the region. Records of SST reconstruction in the BUS during the Pliocene-Pleistocene time intervals (Marlow et al., 2000; Schefuß et al., 2004; Etourneau et al., 2009; Rosell-Melé et al., 2014) also support the possible existence of Benguela Niño-like conditions in the region. Rosell-Melé et al. (2014) promoted the presence of a persistent Benguela Niño-like state before 3.5 Ma in the Pliocene due to the existence of warm SSTs (Salzmann et al., 2011) and a more reduced meridional temperature gradient (Fedorov et al., 2010). Instead, the Pliocene-

Pleistocene transition marked the change towards a colder climatic trend that persisted during the entire Pleistocene (Filippelli and Flores, 2009; McClymont et al., 2013). This period in the Benguela region was characterized by higher SST variations linked to 41-kyr obliquity cycles as well as the increase of the meridional thermal gradient, which led to the intensification of the upwelling phases (Christensen and Giraudeau, 2002; Etourneau et al., 2009; Martinez-Garcia et al., 2010; Rosell-Melé et al., 2014) and the alternation of possible Benguela Niño/Niña conditions. The highest amplitude of SST glacial-interglacial

variability was recorded during the Early-Middle Pleistocene Transition (EMPT), a period between 1.4 and 0.4 Ma, during which a switch from 41-kyr to 100-kyr orbital cycle occurred (Berger and Jansen, 1994; Head and Gibbard, 2015; Herbert, 2023). The planktonic foraminiferal dataset for Hole U1575A and U1576A detected phases of southward/northward shifts of the ABF (up to 24°35´ S), within part of the studied stratigraphic sequence corresponding to the EMPT (Fig. 3; Tables S1-S2). Similar results were published by Ufkes and Kroon (2012), who suggested a possible Benguela Niño-induced southward shift

of the ABF front to 21°S within the EMPT, based on the planktonic foraminifera assemblages. Thus, our results align with the interpretation of Ufkes and Kroon (2012) and show that the ABF reached a more southern position (almost 25°S) during the EMPT. Interestingly, our data for Hole U1575A further indicate that between 0.91 and 0.61 Myr, most of the sediment record exhibits an alternation between normal Benguela conditions and only limited SACW intrusions within the BUS (Figs. 3 and 6). Alkenone-based SST reconstruction north of the ABF (Schefuß et al., 2004) revealed a pronounced SST minimum around

0.90 Ma with a slow SST increase until 0.60 Ma. Thus, we can relate the paleoenvironmental settings inferred from the analysis





of the planktonic assemblage (between 0.90 and 0.61 Myr) as indicative of a period of low SST and associated Benguela Niña conditions. The persistence of Benguela Niña-like phases, in turn, limited the southward ABF shifting and the expansions of the warm SACW into the northern Benguela region. The same conclusions cannot be inferred from Hole U1576A due to a lower biostratigraphic resolution achieved for the abovementioned time interval. Conversely, at Site U1576, we recorded

phases of nutrient filaments transported offshore since the Early-Middle Pleistocene, which may reflect intense coastal upwelling events. This would have been possible during glacial stages, when the ABF is located north of the BUS (thus reducing or impeding the intrusions of SACW), and the SE trade winds intensify, enhancing the upwelling intensity (Manabe and Broccoli, 1985; Jansen et al., 1996; Schefuß et al., 2004).

During the last 0.6 Myrs, the southeastern Atlantic Ocean experienced warmer SSTs both in the north (Etourneau et al.,

2009,2010) and south (Petrick et al., 2015) sectors of the BUS, with an increase in the influx of AgC waters within the Benguela region during deglaciation phases (Peeters et al., 2004; Martínez-Méndez et al., 2008; Dickson et al., 2010; Marino et al., 2013; Caley et al., 2014; Petrick et al., 2015). Conversely, the leakage of the AgC in the Atlantic Ocean was limited during glacials (Biastoch et al., 2008; Bard and Rickaby, 2009). Peeters et al. (2004) and Caley et al. (2012, 2014) observed an increase in the abundance of the warm species *Globorotalia menardii* at ODP Site 1087, located in the southern sector of the BUS, near the

Agulhas retroflection area. The high amount of *Globorotalia menardii* was interpreted as reflective of the intrusions of the Agulhas waters in the BUS prior to interglacial maxima (Petrick et al., 2015,2018). Similarly, the youngest part of our record (since 0.61 Ma) at both sites detected inputs of Agulhas eddies in the northern Benguela region (Fig. 3 and 5), which may correspond to the previously described deglaciation events detected in the southern sector. We can, therefore, hypothesize that, during deglaciations, the position of the subpolar front was sufficiently south to allow the ingression of the warm Agulhas

eddies in the BUS. Furthermore, prior to interglacial maxima, the SSTs are still low enough to limit the shift of the ABF southward, allowing the BOC to have sufficient strength to carry the Agulhas eddies to the northernmost area of the BUS.

Interestingly, sediments from Hole U1576A recorded inputs of warm Indian Ocean waters through the Agulhas leakage also in the Early Pleistocene (between 1.58 and 1.30 Ma; Figs. 3 and 4). Previous studies (Franzese et al., 2006; Caley et al., 2012, 2014; Petrick et al., 2015) provided robust records of the Agulhas leakage from 0.5 Ma to present, based on SST, salinity and

foraminiferal assemblage datasets (Petrick et al., 2018). However, limited data were produced before 0.5 Ma in the BUS, with only Caley et al. (2012) extending the studied record to 1.35 Ma. Furthermore, the variations of the ACC from the Pliocene-Pleistocene transition and their effect on the Agulhas leakage are still poorly known (Keany and Kennett, 1972; Hodell et al., 2000; Becquey and Gersonde, 2002; Diekmann and Kuhn, 2002). Kumar Singh and Sinha (2021) attempted to detect the shifts of the ACC spanning the last 2.6 Ma, with no northward movements observed prior 1.2 Ma. Based on this evidence, we suggest

that the Agulhas leakage signal observed in our study between 1.58 and 1.30 Ma may correspond to a period of deglaciation, during which the polar front was not moving northwards, allowing Agulhas eddies to reach the Southeastern Atlantic Ocean.

## 5 Conclusions


In this study, we analyzed planktonic foraminifera assemblages from IODP Expedition 391, Holes U1575A and U1576A, located on the northwestern sector of the Tristan-Gough-Walvis Ridge (TGW) track in the southeastern Atlantic Ocean. The sites are situated in the northern area of the Benguela Upwelling System (BUS) and are influenced by the Benguela Offshore Current (BOC). The principal aim of this research was to investigate changes in the local paleoceanographic conditions since the onset of the Early-Middle Pleistocene transition (EMPT), based on the analysis of the planktonic foraminiferal assemblages

as well as the application of the Agulhas Leakage Efficiency (ALE) Index and the *Globorotalia truncatulinoides* dextral/sinistral coiling ratio:

1) Age investigation based upon the calcareous nannofossils and planktonic foraminiferal content allowed to establish a reliable biostratigraphic framework (covering the Early-Middle to Late Pleistocene) for the paleoecological interpretations of Holes

U1575A-U1576A.

2) Cluster analysis on the planktonic foraminifera assemblages produced 3 main clusters for Hole U1575A and 2 major clusters for Hole U1576A, with cluster 1 subdivided into two subclusters (1a and 1b) at both sites. Overall, those clusters represent different paleoceanographic conditions during the Pleistocene, reflecting phases of northward/southward shifts of the ABF,

normal BOC conditions, increased offshore nutrient transports and intrusions of the Agulhas waters from the Indian Ocean. The paleoecological interpretation of the clusters is further supported by PCA results.

3) The northward/southward migrations of the ABF were detected based on the abundance of *Globorotalia crassaformis* compared to *Globorotalia inflata*. Specifically, a concomitant increase of *Globorotalia crassaformis* and a reduction in

abundance of *Globoconella inflata* indicated a southward shift of the ABF and consequent ingressions of SACW within the northern sector of the BUS. Contrarily, normal BOC conditions with no SACW mixing was corroborated by higher values of *Globoconella inflat*a and low abundances of *Globorotalia crassaformis*.

4) *Globorotalia truncatulinoides* dextral/sinistral coiling ratio and their abundance data reflected changes in the stability of the

thermocline, temperature and productivity at the studied sites. During phases of southward movements of the ABF, a higher increase in abundance of *Globorotalia truncatulinoides* sinistral compared to the dextral, as well as a negative coiling ratio, were indicative of reduced water mixing between the BOC and the SACW, causing a slight decrease in the thermocline stability and a rise in water temperature. Conversely, similar abundances of *Globorotalia truncatulinoides* dextral and sinistral and a more positive value of the coiling ratio suggested a stronger water mixing (leading to a stronger instability of the local

thermocline than what observed in cluster 1b) and higher increase in water temperature. Periods of normal BOC conditions, with no intrusions of the SACW in the BUS, showed the presence of a relatively stable thermocline with colder water temperature, which favored both coiling types. Furthermore, in Hole U1576A, the higher abundance of *Globorotalia truncatulinoides* sinistral was also interpreted as indicative of less fertile waters within the BOC.

5) The local thermocline response is strictly related to longitudinal and latitudinal variations of the oceanographic conditions in the studied area. This can be seen in Hole U1576A which is situated in a more southern position and closer to the center of the gyre, compared to Hole U1575A. Thus, we observed only a subtle thermocline variability as the warm SACWs/Agulhas eddies mix with already more temperate waters within the BUS.

6) The ALE Index together with the analyses of the planktonic foraminifera assemblage and the *Globorotalia truncatulinoides* coiling ratio support the interpretation that *Globorotalia menardii* and the other tropical taxa represent a reseeding population from the Indian Ocean, associated with the Agulhas leakage. Higher (lower) values of the ALE Index indicate enhanced (reduced) transport of Indian Ocean warm waters within the BUS. This study further proved the reliability of the ALE Index to successfully trace the input of the Agulhas waters in the northernmost sector of the BUS.


7) A sharp increase in the abundance of *Orbulina universa* in Hole U1576A indicates the existence of temperate and moderately nutrient-enriched waters, which can be explained by the transport of nutrient filaments offshore, far from the coastal upwelling area.

8) The planktonic foraminifera dataset at both sites recorded phases of ABF shift and periods of increase in the Agulhas leakage from the EMPT to the recent. Southwards/northwards movements of the ABF were interpreted to reflect long-term changes in the occurrence of Benguela Niño/Niña events in the SE Atlantic Ocean. Particularly, between 0.91 and 0.61 Myr, we detected intervals of possible Niña-like conditions in the northern BUS, corresponding to periods of low SSTs recorded in the region from previous studies. Instead, phases of Agulhas water ingressions in the BUS may be associated with deglaciation stages

since the onset of the EMPT.

**Data availability**

The oceanographic map was made with Ocean Data View (Schlitzer, 2021) version 5.6.2, available at https://odv.awi.de.

Statistical and ordination analyses were performed using PAST (version 4.09) (Hammer et al., 2001), available at https://www.nhm.uio.no/english/research/resources/past/.

The quantitative data that support the findings of this study are openly available in PANGAEA at https://www.pangaea.de/.

**Competing interests**


The contact author has declared that none of the authors has any competing interests.





**Acknowledgments**

Samples and data provided by the International Ocean Discovery Program (IODP) were used for this research project. The authors would like to express their gratitude to all the personnel of the D/V JOIDES Resolution for their work during IODP Expedition 391 as well as the shipboard science party for collecting the shipboard data. All shipboard data are publicly available from www.iodp.tamu.edu. AVDG warmly thanks the Austrian Academy of Science (ÖAW) for providing fundings to support her participation in the IODP Expedition 391 as shipboard scientist. This study was funded by the Austrian Science

Fund (FWF) (project number P 31683-N29).




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

**Appendices**

**Appendix A:** Plate 1 showing the relevant calcareous nannofossil species used as biostratigraphic indicators in Holes U1575A and U1576A.

**Appendix B:** Plates 2-6 illustrating the main planktonic foraminifera species used for biostratigraphic and paleoceanographic interpretations of Holes U1575A and U1576A



# Plate 1





**Plate 1**. **1-***Calcidiscus macintyrei*, Sample U1575A-5R-7W, 0-2 cm. (a) LM, X nicols; (b) LM, FC; (c) LM, quartz wedge interference image. **2-***Helicosphaera sellii*, Sample U1575A-5R-3W, 0-2 cm. LM, X nicols. **3-***Discoaster asymmetricus*, Sample U1575A-5R-7W, 0-2 cm. LM, X nicols. **4-***Pseudoemiliania lacunosa*, Sample U1576A-3R-3W, 145-147 cm. LM, X nicols. **5-6** *Reticulofenestra asanoi*, Sample U1575A-3R-6W, 0-2 cm. LM, X nicols. **7-**small *Gephyrocapsa* spp. (<4 µm), Sample U1575A-1R-2W, 0-2 cm. a) LM, X nicols; (b) LM, FC. **8-**medium *Gephyrocapsa* spp. (4-4.5 µm), Sample U1575A-1R-1W 0-2 cm. a) LM, X nicols; (b) LM, FC. **9-**large *Gephyrocapsa* spp.(>5.5 µm), Sample U1575A-4R-3W, 88-90 cm. (a) LM, X nicols; (b) LM, FC. **10-11** *Discoaster brouweri*, Sample U1576A-4R-3W, 0-2 cm. LM, FC. **12-13** *Discoaster triradiatus*, Sample U1576A-4R-3W, 0-2 cm. LM, FC. **14-15** *Emiliania huxleyi*, Sample U1576A-1R-1W, 0-2 cm. SEM.



# Plate 2





**Plate 2**. **1**-*Globoturborotalita rubescens*, Sample U1576A-1R-3W, 60-62 cm. **2**-*Trilobatus sacculifer*, Sample U1576A-1R-3W, 60-62 cm. (a) umbilical view; (b) spiral view. **3**-*Neogloboquadrina pachyderma*, Sample U1576A-1R-3W, 60-62 cm. (a) umbilical view; (b) spiral view. **4**-*Neogloboquadrina incompta*, Sample U1576A-1R-3W, 60-62 cm. umbilical view. **5**-*Neogloboquadrina incompta*, Sample U1576A-1R-3W, 60-62 cm. (a) umbilical view; (b) spiral view. **6**-*Globoconella inflata*,

Sample U1576A-1R-7W, 58-60 cm. (a) umbilical view; (b) spiral view. **7**-*Globoconella inflata*, Sample U1576A-1R-7W, 58-60 cm. umbilical view. **8**-*Globoconella inflata*, Sample U1576A-3R-6W, 10-12 cm. side view. **9**-*Globigerinita glutinata*, Sample U1576A-1R-3W, 60-62 cm. (a) umbilical view; (b) side view; (c) spiral view. **10**-*Orbulina universa*, Sample U1576A-1R-3W, 60-62 cm. All specimens were imaged at 3kv and with external secondary electron (SE) detector.










# Plate 3





**Plate 3. 1**-*Globigerinoides ruber* s.s., Sample U1576A-1R-3W, 60-62 cm. (a) umbilical view; (b) side view; (c) spiral view. **2**-*Globigerinella siphonifera*, Sample U1576A-3R-6W, 10-12 cm. (a) umbilical view; (b) side view. **3**-*Globigerinoides ruber* s.l., Sample U1576A-1R-3W, 60-62 cm. (a) umbilical view; (b) side view; (c) spiral view. **4**-*Globigerina bulloides*, Sample U1576A-1R-3W, 60-62 cm. (a) umbilical view; (b) side view; (c) spiral view. **5**-*Sphaeroidinella dehiscens*, Sample U1576A-1R-3W, 60-62 cm. (a) umbilical view; (b) spiral view. **6**-*Turborotalita quinqueloba*, Sample U1576A-3R-7W, 0-2 cm. (a) umbilical view; (b) side view; (c) spiral view. All specimens were imaged at 3kv and with external secondary electron (SE) detector.



# Plate 4





**Plate 4. 1**-*Globorotalia menardii*, Sample U1575A-3R-1W, 0-2 cm. (a) umbilical view; (b) side view; (c) spiral view. **2**-*Globorotalia crassaformis*, Sample U1575A-3R-1W, 0-2 cm. (a) umbilical view; (b) side view; (c) spiral view. **3**-*Globorotalia hirsuta*, Sample U1576A-1R-3W, 60-62 cm. (a) umbilical view; (b) side view; (c) spiral view. **4**-*Globorotalia truncatulinoides* (destral), Sample U1575A-1R-1W, 140-142 cm. (a) umbilical view; (b) side view. **5**-*Globorotalia truncatulinoides* (dextral), Sample U1575A-3R-4W cm, 138-140. spiral view. **6**-*Globorotalia truncatulinoides* (sinistral), Sample U1576A-1R-3W, 60-

62 cm. (a) umbilical view; (b) side view; (c) spiral view. **7**-*Globorotalia tosaensis*, Sample U1576A-3R-7W, 0-2 cm. umbilical view. All specimens were imaged at 3kv and with external secondary electron (SE) detector.






# Plate 5



**Plate 5. 1**-*Globorotalia tosaensis*, Sample U1576A-3R-7W, 0-2 cm. (a) side view; (b) spiral view. **2**-*Globorotalia hessi*, Sample U1575A-1R-4W, 10-12 cm. (a) umbilical view; (b) side view; (c) spiral view. **3**-*Globorotalia flexuosa*, Sample U1576A-2R-1W, 0-2 cm. (a) umbilical view; (b) side view; (c) spiral view. **4**-*Neogloboquadrina acostaensis*, Sample U1575A-5R-4W, 135-137 cm. (a) umbilical view; (b) side view; (c) spiral view. **5**-*Globigerinella calida*, Sample U1576A-1R-1W, 0-2 cm. (a) umbilical view; (b) spiral view. **6**-*Dentoglobigerina altispira*, Sample U1575A-5R-7W, 0-2 cm. (a) umbilical view; (b) side view. (c) spiral view.  All specimens were imaged at 3kv and with external secondary electron (SE) detector.





# Plate 6



**Plate 6. 1**-*Globigerinoides obliquus*, Sample U1576A-4R-2W, 10-12 cm. (a) umbilical view; (b) side view; (c) spiral view. **2**-*Globigerinoides extremus*, Sample U1576A-4R-3W, 0-2 cm. (a) umbilical view; (b) side view; (c) spiral view.

1480  All specimens were imaged at 3kv and with external secondary electron (SE) detector.