# Peer review of "Planktonic foraminiferal assemblages as tracers of paleoceanographic changes within the Northern Benguela current system since the Early Pleistocene"

_Climate of the Past, 2024_

## Author Response (AR1)

**Arianna Valentina Del Gaudio**

*Institute for Earth Sciences, NAWI Graz Geocenter,*
*University of Graz*
*Heinrichstraße 26, 8010 Graz, Austria*
E-mail: arianna.del-gaudio@uni-graz.at

To

**Erin McClymont**

Editor
Climate of the Past

Graz, 4^th August 2024

Dear Editor,

On behalf of my co-authors, I would like to thank you for the opportunity to revise and resubmit our manuscript entitled "Planktonic foraminiferal assemblages as tracers of paleoceanographic changes within the Northern Benguela current system since the Early Pleistocene".
We are grateful to you and the reviewers for their insightful comments on our manuscript. We have been able to incorporate changes to reflect most of the suggestions provided by the reviewers and the editor.
Our responses to the reviewers and editor, are prefaced as "Authors´ response". Corresponding changes are highlighted in the manuscript text in the revised file.
We hope that these revisions are sufficient to make our manuscript suitable for publication in *Climate of the Past*.

Sincerely,

Arianna Del Gaudio Ph.D.
*Institute for Earth Sciences, NAWI Graz Geocenter*
*University of Graz*

**Response to the Editor**

**The manuscript received two reviews and there was discussion regarding the biostratigraphy and the statistics with reviewer 2. In general, the authors have been able to respond to the comments and suggestions made by the reviewers and have indicated where changes can be made to the manuscript to address these concerns.**

**I want to address the exchange between Reviewer 2's comments on time frames and the descriptive nature of the text, which the authors described as "inflammatory and destructive comments". Having read all of the reviewers comments and returned to the submitted manuscript, I sympathise with Reviewer 2 in their request for needing a stronger sense of time on the conceptual diagram but also via comparison of the new data with published records. Although it may be clear to the authors what the different age zonations mean on their graphics of down-core data, I see two issues:**

**First, section 4.2 is called "paleoceanographic evolution...." but this is really a description of the interpretation of the different clusters and the processes which might explain them, but it is not set into any time context. When did these changes occur? How do they align with evidence from other sites? Figure 6 does not say when these clusters occurred: early or late in the record? Sporadically throughout? Glacial or interglacial? Here is where I think Reviewer 2 identifies the "decriptive" nature of the text, because it says that things occurred, but not when.**

**Second, section 4.3 is titled "variations of the paleoceanographic conditions..." but the first two paragraphs have little information about when these occurred, until line 677 when the authors flag a time window. If the authors feel that the repeated appearance of the clusters mean that there is no clear long-term trend but rather a variability over time on shorter timescales then highlighting this at the start of 4.2 or 4.3 would be worthwhile.**

**Finally, I wonder if some of the concerns raised by Reviewer 2 about the palaeoceanographic interest and time-evolution of the records could be addressed if the two sites and their data could be integrated over time, which can likely be addressed through edits to section 4.3 (see below). The two sites are still kept very separate (e.g. see comment below about Figure 6) but by combining the information they could reveal a quite detailed assessment of the regional evolution of the oceanography.**

Authors´ response: We gladly thank the editor and both the reviewers for the insightful comments on our manuscript that led to possible improvements in the current version. All the comments raised by the reviewers and the editor were carefully taken into consideration and we present our reply to them in the following paragraphs. This first part of the response includes our answer to the general observations from the editor whereas the second part (see below) focuses on answering the specific comments indicated by the reviewers and the editor. We agree that using the terminology "paleoceanographic evolution" for the title of section 4.2 may create confusion. Specifically, as the section 4.2 highlights the paleoceanographic conditions based on the planktonic assemblages but without any temporal context, we decided to modify the title of the section 4.2 to "Paleoceanographic conditions in the northern Benguela system inferred from the planktonic foraminiferal assemblages" (Please refer to lines 503-504 of the revised manuscript). Consequently, the title for section 4.3 was changed to "Paleoceanographic evolution of the northern BUS during the Quaternary" as here the regional paleoceanographic evolution (based on the clusters defined in chapter 4.2) through time is discussed (Please refer to lines 667-668 of the revised manuscript). In order to solve the issues related to the timing when the paleoceanographic changes occurred and on which temporal scale, we modified the first paragraph of chapter 4.3 as follows: "The planktonic foraminiferal records in Holes U1575A and U1576A indicate large scale variability in the paleoenvironmental conditions since the Early Pleistocene (Figs. 6-7). Specifically, we observed an alternation of periods reflective of variations

in the ABF positions, Agulhas intrusions from the Indian Ocean and an increase of the nutrient transport offshore, further away from the coastal upwelling center. Those paleoceanographic conditions were sporadically recorded within the whole analyzed time interval (Figure 7)." Please refer to lines 670-674 of the revised manuscript.

Finally, to emphasize the palaeoceanographic interest and time-evolution of the records, the results from the two sites were integrated over time and compared with previous abundance and SST records from the region (see chapter 4.3).
* * *
**Unless otherwise stated, the proposed edits in response to the reviewers should be incorporated into a revised text alongside consideration of the Editor comments below:**

**1) Section 4.1: in addition to the suggested corrections offered by the authors in response to the discussion with Reviewer 1, it would be helpful to have a short explanation of the differences in the "IODP" approach and the one described here, for those readers who are less familiar with the methods. I suggest that in the introduction to Section 4.1 the authors should briefly flag that as the sediment was not sampled between biostratigraphy horizons there is uncertainty in the actual mbsf of that biostratigraphic event (they can use a compressed version of what is written in response to reviewer 1). Both the authors and the reviewer have valid concerns about the presentation of the datums, so having the approach and its implications clearly outlined here would likely be useful for other readers in future.**

Authors´ response: we are thankful to the editor for the comments. We have inserted a paragraph on the use and how to calculate the depth error in the methodology section as follows:

"We calculated a depth error value for each biostratigraphic event (Tables 1 and 2), which represent the uncertainty of the depth at which a specific biohorizon (species' base/top) occurs within the sequence. Specifically, the maximum potential depth error for the planktonic foraminifera and calcareous nannofossil datums is expressed as the sample spacing between the sample in which the specific bioevent was placed and the stratigraphically next sample (for the top occurrence) /previous sample (for the base occurrence)." Please refer to lines 224-229 of the revised manuscript.

We also added the explanation about why we prefer the maximum depth error approach in section 4.1 as follows:

"Sediment samples used for this study were not collected within the cores between biostratigraphic horizons, leading to an uncertainty in the placement of the biostratigraphic events. To evaluate the degree of this uncertainty for the bioevents, we calculated the maximum potential depth error (see section 2.4 in Materials and Methods) rather than the mid-point depth approach (commonly used in IODP data reporting) where a bioevent is placed as midpoint between the sample in which a taxon is first (or last) recorded and the sample stratigraphically below (or above) within which the taxon is not present. We deliberately chose this approach because the use of the mid-point depths obscures critical information on the potential sampling bias in depth direction (Top and Base occurrences can logically only have an error up and downward section, respectively). Thus, the use of the maximum potential depth error has the benefit that its degree of freedom is, per definition, unidirectional for each event. Moreover, we can add additional and highly valuable information and details to the generated age-depth model". Please refer to lines 405-414 of the revised manuscript.

**2) Figures 3 and 5 (cluster colours, raised by Reviewer 1): the confusion here may come from the similarity in the green/blue between cluster 1b (U1575) and cluster 2 (U1576A). Is it possible to adjust one of them so that there is easier differentiation between the two colours ?**

Authors´ response: We thank the reviewer for the observation. The color for cluster 1b (Hole U1575A) was changed from blue to purple. We subsequently also desaturated the orange color for cluster 2 to a more yellowish tone to make the graphs looking more harmonic. Changes have applied to Figures 3, 4, 5, 6, 7 and Supplementary file Table S3.

**3) Comment 4 by reviewer 1: the authors indicate some uncertainty about the reviewer request to correlate conditions between the two sites. Perhaps what was intended was to put the combined data from U1575 and U1576 into a single time-series and cluster analysis, so that you identify the broader regional changes over time, rather than moving back and forth across the records from two sites? Are the oceanographic conditions at the two sites similar enough for this to be achieved, or do you expect that there are strong enough local differences for the sites to stay separate?**

Authors´ response: we are thankful to the editor for the suggestions. In this work we wanted to highlight the possible differences in the ecological and paleoceanographic conditions in two different portions of the northern BUS. From the cluster and PCA analyses we saw that, despite some clusters define similar paleoceanographic conditions in the region, the local ecological and paleoceanographic response to those conditions was different in the two portions of the BUS. Thus, we think that it is vital to keep the two sites separated to highlight the latitudinal variations under slightly similar paleoceanographic conditions. However, we attempted to produce a single cluster analysis including both sites, as suggested by the editor (see figures below).

UPGMA (Bray-Curtis)

[Figure]

WARD (Euclidean)

[Figure]

This approach, however, did not provide senseful results, as all the samples linked to the recorded paleoceanographic conditions showed a random distribution which cannot be interpreted. Specifically, the cluster of increase nutrient filaments offshore (with high *O. universa* content) is completely mixed with the expanded and limited SACW ingression ones. Neither *G. crassaformis* or *G. inflata* seem to be the reason for the clustering. In fact, the amount of *G. inflata* in the samples with *O. universa* content is higher than *G. crassaformis* but those samples cluster with the expanded/limited SACW expansion samples (showing higher *G. crassaformis* content) rather than the ones with normal BOC conditions (higher *G. inflata*). Also, all the information that we obtained with PCA analysis seems to be lost. As expanded/limited samples are also randomly mixed-up, the effect of *G. truncatulinoides* dextral and sinistral cannot be detected anymore. Most importantly, having a unique clustering for both sites can obscure the effect of the latitude and, thus, the mixing of limited and expanded conditions does not allow to clearly show the evolution of the ABF through time (as those differences are linked to the latitudinal position of the sites).

We want to highlight that, even if we kept the two sites separated, it was still possible to reconstruct the regional paleoceanographic evolution through time, by linking the different conditions of the two sites during the same time intervals (see chapter 4.3). This, in turn, provides a more complete paleoceanographic history because we can define what happened in different positions of the BUS during the Pleistocene.

**4) Related comment: the authors suggest in response to reviewer 1 that they could make an additional conceptual model for U1576: given that figure 6 includes an east-west component, is it not possible to add the position of U1576 onto this conceptual diagram and flag which of the U1576 clusters might align with the ones shown for U1575?**

Authors´ response: we are thankful to the editor for the comment. As also discussed with reviewer 1, we included the position of Hole U1576A in Figure 6. Specifically, the position of Hole U1576A was added to the conceptual model (Figure 6). Regarding the request to flag the clusters of the two sites, we indicated the clusters associated with each paleoceanographic condition for both sites U1575 and 76 in Figure 6 and related caption (Please refer to lines 569-572 of the revised manuscript).

**5) Comment 4 by reviewer 1: the authors suggest that they can't make comparisons to Ufkes and Kroon (2012) as the latter only extend to 1.1 Ma. But this still gives several hundred thousand years of overlap, where some comparisons could be made e.g. calculating %ALE, or identifying relative abundance of key species (e.g. G crassaformis, as outlined in e.g. line 498?), perhaps in combination with edits to Section 4.3 (see below).**

Authors´ response: we kindly thank the editor for the suggestion. We made comparison of our data with the works from Ufkes and Kroon (2012), Caley et al. (2012) in combination with the SST records of Schefuß et al. (2004), Etorneau et al. (2009), Petrick et al. (2015) as suggested in order to describe the regional evolution of the northern BUS oceanographic conditions with time (see chapter 4.3). Specifically, we plotted the relative % of *G. menardii* and *G. crassaformis* and we calculated and plotted the ALE index values using the dataset of Ufkes and Kroon (2012).

**6) Section 4.2: in the introduction to this section, it would be helpful to highlight the average time resolution of sampling, which will show that orbital-scale analysis isn't resolved but broad trends can be investigated. This will help caveat author concerns about their resolution being different to the existing SST data, but also gives the reader a sense of what to expect from the data. This is also what Reviewer 2 was suggesting.**

Authors´ response: we are grateful for this observation made by the editor and reviewer 2. We inserted the average time resolution of the sampling for both Holes at the beginning of chapter 4.2. as follows: "Variations of the planktonic foraminiferal assemblages during the Quaternary can be interpreted as indicative of changes in the water mass dynamics within the northern Benguela system. In this respect, quantitative analyses performed on the foraminiferal communities (UPGMA and PCA) revealed different paleoceanographic settings in Holes U1575A and U1576A, with an average time resolution for the sampling between 50 (Hole U1575A) and 70 Kyr (U1576A)." Please refer to lines 506-511 of the revised manuscript.

**7) Section 4.3: I have sympathy with Reviewer 2 describing some of this discussion as descriptive, because in the early part of this section (paragraph starting line 643) there are "shifts" described but at the end of the paragraph it's not clear whether these were long-term or short-term , and how they connect with events in other records showing e.g. evolution of BUS or ABF over time. T here is quite a bit of text about Benguela Nino-like state before 3.5 Ma (e.g. line 662) but that time window doesn't overlap with this study, so is it relevant? The text which follows keeps referring to "within the EMPT" but what does this mean? It is only from line 677 that dates are provided, so I'm unclear whether this refers to the preceding text or the text which follows. From line 689 the links with other records during the time interval of interest is much better achieved.**

Authors´ response: we thank the reviewer 2 and the editor for these suggestions. We extensively modified chapter 4.3 to accommodate all the requests stated in this comment as follows:

- We have indicated that we are looking at short term shifts. In this respect, we added the following paragraph (please refer to lines 670-671 of the revised manuscript): "The planktonic foraminiferal records in Holes U1575A and U1576A indicate large scale variability in the paleoenvironmental conditions since the Early Pleistocene (Figs. 6-7)."

- We connected our results with other records (Ufkes and Kroon, 2012; Caley et al. 2012) and SST records (Schefuß et al. 2004; Etourneau et al. 2009 and Petrick et al. 2015) to better reconstruct the paleoceanographic evolution of the BUS over time (see chapter 4.3).

- The large amount of text related to the Pliocene-Pleistocene climatic shift interval was removed (please refer to lines 696-702 of the revised manuscript) and the paragraph related to the EMPT was modified accordingly, to better link it with the previous paragraph (please refer to lines 702-704 of the revised manuscript).

- All the discussed time intervals (including the EMPT interval) for the paleoceanographic conditions are now clearly defined in the text (see chapter 4.3).

**8) Reviewer 2 queries why some of the published data cited in Section 4.3 isn't shown. I agree that this makes it hard to compare the outcomes of this new work with the regional context, although I also acknowledge the issues of differing time resolution. Given that the sampling here seems to be ~25 kyr , I suggest plotting one or two key records with a smoothing at that value (or longer period) so that readers eyes are drawn to longer-term trends and how this new data sits in that context. It doesn't need to be lots of SST data, it could be a key SST record, or a SST gradient between sites, or a couple of key foraminifera % records which give a regional context. Here is where I think you could address Reviewer 2's concerns about providing a history of the regional circulation system with a clear graphic of selected published material alongside selected new data from this work.**

Authors´ response: we are grateful for the suggestions made by the editor and reviewer 2 which really helped us improve the outcomes of our manuscript. We plotted our data together with the abundance data and ALE index from Ufkes and Kroon, 2012, Caley et al. 2012, as well as the SST records (Schefuß et al. 2004; Etourneau et al. 2009 and Petrick et al. 2015) as requested (Figure 7). The dataset used to create Figure 7 has also been included as Table S7.  Thanks to this data comparison, we not only better provided and displayed the history of the regional circulation system but also added new information on the paleoceanography of the region which are now part of the discussion (see chapter 4.3).

**9) In the reply to reviewer 2, the authors state that "we managed to detect important variations in the paleoceanographic conditions (thermocline fluctuations) in a very complex oceanic region, as well as to characterize the main water masses (Benguela Oceanic Current, South Atlantic Central Waters, and Indian Ocean water influxes), which exert an influence in the Benguela Upwelling System. Moreover, we statistically demonstrated that the increase in G. menardii within the BUS is related to the Agulhas Leakage and not to the variations in the depth of the thermocline". I do not find these findings to be clearly stated in section 4.3, where the palaeoceanography is described: for example, "thermocline" is not used at all in section 4.3, and I find it hard to determine from that section (or 4.2) when the variations that the authors identify as key took place  (noting also Reviewer 2's concerns that the data is only ever plotted by depth ).**

Authors´ response: we are grateful for the observations made by the editor. We now included the thermocline variations associated with the main paleoceanographic conditions also in chapter 4.3 to provide to the readers the complete paleoceanographic history of the northern BUS region.
We also specified that our work clearly indicated that *G. menardii* clearly belong to the Indian Ocean domain in chapter 4.3 (please refer to lines 660-661 of the revised manuscript).
* * *
Finally, we want to highlight that we incorporated all the suggestions made by reviewer 1 mentioned during the public discussion phase. Specifically, we summarized here the major changes as followed:

- We provided in our two tables an error value for each biostratigraphic event, defined as the uncertainty of the depth at which a specific biohorizon (species' base/top) occurs within the sequence.

- The bioevent T *D. altispira* has been removed from the list of bioevents. In this respect, we changed the paragraph "The oldest detected bioevent at Hole U1575A is the top occurrence 395 (T) of *Dentoglobigerina altispira* (Sample U1575A-5R-7W, 0-2 cm; 46.89 mbsf), which was recorded in the Atlantic Ocean at 3.13 Ma (Wade et al., 2011; Gradstein et al., 2020). The abovementioned sample also contains the base (B) of the planktonic taxon *Globoconella inflata* (B 3.24 Ma; Gradstein et al., 2020)." to: "The oldest possible age interval recorded at Hole U1575A (Sample U1575A-5R-7W, 0-2 cm; 46.89 mbsf) is between 3.13 and 3.24 Ma, based on the top occurrence (T) of *Dentoglobigerina altispira* (Wade et al., 2011; Gradstein et al., 2020) and the base (B) of the planktonic taxon *Globoconella inflata* (Gradstein et al., 2020), respectively." (Please refer to lines 418-422 of the revised manuscript).

- We also decided to change the sentence "Bioevents were defined using base (B) and top (T) as well as the base common (Bc) and top common (Tc) occurrences of marker taxa (see Tables S1-S2 and Tables 1 and 2)." to "Local bioevents were defined using base (B) and top (T) as well as the base common (Bc) and top common (Tc) occurrences of marker taxa (see Tables S1-S2 and Tables 1 and 2)." Please refer to lines 223-224 of the revised manuscript. This will clarify that we defined the occurrences of the taxa within the studied stratigraphic sequence, with the ages based on the most recent compilation in the Geologic Time Scale 2020.

- We inserted the paragraph about the existence of a possible hiatus at Site U1575 as followed: "A possible hiatus can be placed between 44.20 and 46.89 mbsf, as an abrupt change in the sedimentation rate can be observed within this interval. However, the presence of the hiatus cannot be confirmed with certainty due to the lack of available samples between sections 4W and 7W in Core 5H". (Please refer to lines 426-429 of the revised manuscript).

- We added the paleoceanographic information associated with all the clusters in the figure 3 and 5 captions as followed: "Clusters for Hole U1575A are indicated as follows: 1a=Expanded SACW intrusion; 1b=Limited SACW intrusion; 2=Normal BOC conditions; 3=Agulhas Leakage. Clusters for Hole U1576A are indicated as follows: 1a=Expanded SACW intrusion; 1b=Agulhas Leakage; 2=Nutrient filaments within the BOC." Please refer to lines 304-309 and 362-366 of the revised manuscript.

- Site U1576 was integrated in the conceptual model (Figure 6) to show the main paleoceanographic conditions at this location.

- Agulhas leakage sketch was added in Figure 6.

- We changed "Discoaster" at line 287 of the original manuscript to italics (refer to line 290 of the revised manuscript).

- We changed the sentence "A total of 6996 and 5540 planktonic foraminiferal specimens were identified for the assemblage study in Holes U1575A and U1576A, respectively." with "An average of 304 (min=283; max=327) and 308 (min=297; max=346) specimens per sample were counted and identified for the assemblage study in Holes U1575A and U1576A, respectively". refer to lines 297-298 of the revised manuscript.

- We could not substitute Hole with Site as, according to the IODP terminology, the expression "Holes U1575A and U1576A" is the correct option. However, we noted that there is a plural form missing at line 672 (now line 711 of the revised manuscript). Thus, we changed the term "Hole" with "Holes".
* * *
**Additional clarifications**

We also want to inform the editor that, in addition to the above comments, all spelling, grammatical mistakes and missing italics font for the scientific names of the species have been checked and, if wrong, corrected.

Furthermore, we checked all the references and modified them to include or cancel the references linked to chapter 4.3 which was extensively modified.